# Inducible cell-specific mouse models for paired epigenetic and transcriptomic studies of microglia and astroglia

Ana J. Chucair-Elliott [1,9], Sarah R. Ocañas [1,2,9], David R. Stanford[1], Victor A. Ansere[1,2], Kyla B. Buettner [1,2], Hunter Porter[1,3], Nicole L. Eliason[4], Justin J. Reid[5], Amanda L. Sharpe [4], Michael B. Stout[6], Michael J. Beckstead [5], Benjamin F. Miller[5], Arlan Richardson[7,8] & Willard M. Freeman [1,7,8 ✉]

Epigenetic regulation of gene expression occurs in a cell type-specific manner. Current cell-type specific neuroepigenetic studies rely on cell sorting methods that can alter cell phenotype and introduce potential confounds. Here we demonstrate and validate a Nuclear Tagging and Translating Ribosome Affinity Purification (NuTRAP) approach for temporally controlled labeling and isolation of ribosomes and nuclei, and thus RNA and DNA, from specific central nervous system cell types. Analysis of gene expression and DNA modifications in astrocytes or microglia from the same animal demonstrates differential usage of DNA methylation and hydroxymethylation in CpG and non-CpG contexts that corresponds to cell type-specific gene expression. Application of this approach in LPS treated mice uncovers microglia-specific transcriptome and epigenome changes in inflammatory pathways that cannot be detected with tissue-level analysis. The NuTRAP model and the validation approaches presented can be applied to any brain cell type for which a cell type-specific cre is available.

[1] Genes & Human Disease Program, Oklahoma Medical Research Foundation, Oklahoma City, OK, USA. [2] Department of Physiology, University of Oklahoma Health Sciences Center, Oklahoma City, OK, USA. [3] Oklahoma Center for Neuroscience, University of Oklahoma Health Sciences Center, Oklahoma City, OK, USA. [4] Department of Pharmaceutical Sciences, University of Oklahoma Health Sciences Center, Oklahoma City, OK, USA. [5] Aging & Metabolism Program, Oklahoma Medical Research Foundation, Oklahoma City, OK, USA. [6] Department of Nutritional Sciences, University of Oklahoma Health Sciences Center, Oklahoma City, OK, USA. [7] Department of Biochemistry, University of Oklahoma Health Sciences Center, Oklahoma City, OK, USA. [8] Oklahoma City Veterans Affairs Medical Center, Oklahoma City, OK, USA. [9] These authors contributed equally: Ana J. Chucair-Elliott, Sarah R. Ocañas. ✉email: bill-freeman@omrf.org

Considerable advances are being made in understanding the epigenome and its relationship with gene expression in the brain[1–3]. However, the lack of approaches for paired analysis of DNA and RNA profiles at the cell-type-specific level within the same animal is a limitation for the field, given that epigenetic processes differ across central nervous system (CNS) cell types at the level of chromatin organization and DNA modifications[1,4]. Obtaining enriched cell populations by flow sorting requires cell surface markers, these markers can change with experimental conditions, and cell sorting causes molecular, morphological, and functional changes, such as cell activation, that could confound studies[3,5,6]. Single-cell approaches[7] may overcome some of the challenges of cell sorting but the scale of such studies, incomplete genomic coverage, restriction to only certain types of endpoints, and continued potential for brain dissociation artifacts are limitations.

This has led to development of transgenic labeling approaches to isolate RNA or DNA from specific cell types. Ribosome labeling and RNA isolation methods, such as Translating Ribosome Affinity Purification (TRAP[8]), and ribosome tagging (RiboTag[9]), are gaining acceptance across neuroscience studies examining the transcriptome. Similar approaches have been developed to transgenically tag and allow isolation of nuclei and thus DNA (Isolation of Nuclei TAgged in specific Cell Types, INTACT)[10]. However, using separate transgenic mouse strains for DNA and RNA endpoints is a complicated and resource intensive approach.

The Nuclear Tagging and Translating Ribosome Affinity Purification (NuTRAP) system was first applied by Roh et al.[11] in the parallel, cell-specific isolation and characterization of mRNA expression and chromatin states from adipocyte and hepatocyte populations in vivo, upon Adiponectin-Cre- or Albumin-Cre-dependent recombination[11]. Here, the NuTRAP construct is combined with well-established cell-specific inducible cre-recombinase expressing systems[12,13] to perform paired transcriptomic and epigenomic analyses (gene expression and DNA modifications) of specific CNS glial cell types, astrocytes and microglia in a temporally controllable manner from a single mouse. Demonstration studies provide: (1) cell-type-specific enrichment of RNA and DNA, (2) novel insights into differential usage of DNA modifications in microglia and astrocytes, and (3) examples of cell-type-specific transcriptomic and epigenomic responses that are only revealed when specific cell types are examined. These studies also provide a validation approach NuTRAP mouse lines crossed to any available cre driver line relevant to neuroscience studies.

## Results
Schematics of the NuTRAP construct, experimental design, and key protocols for the analyses in the current study are represented in Fig. 1. Of note, the Aldh1l1-cre/ERT2; NuTRAP and Cx3cr1-cre/ERT2; NuTRAP models will be abbreviated when necessary, as Aldh1l1-NuTRAP and Cx3cr1-NuTRAP, respectively. Testing of Tamoxifen (Tam) administration, for cre induction, effects on the epigenome and transcriptome in the CNS found no long-lasting, significant effects on DNA modifications or gene expression[14].

**Flow cytometry and immunohistochemical validation of the Aldh1l1-cre/ERT2+; NuTRAP+ mouse brain**. The Aldh1l1-cre/ERT2+ mouse line has been reported as highly efficient and specific for DNA recombination in astrocytes[12]. We first crossed this line with the NuTRAP reporter mouse[11] to couple epigenomic and gene expression studies in astrocytes in a parallel fashion. As an initial validation of the model, Aldh1l1-NuTRAP

mice were systemically delivered Tam for 5 consecutive days and a week after induction, brains were dissected for flow cytometry (FC) and immunohistochemistry (IHC) analyses. Single-cell suspensions of brains immunostained with ACSA-2 antibody, a pan-astrocytic marker[15], revealed a distinct EGFP+ cell population present in the Aldh1l1-NuTRAP brains but not in the cre-negative counterparts, consistent with the reported 10–20% astroglial cellularity in the rodent brain[16,17]. Almost the entirety of the EGFP+ cell population co-expressed ACSA-2, supporting that cre-mediated recombination upon Tam induction specifically targeted astrocytes (Fig. 2a, b).

Sagittal brain sections immunostained with cell-specific markers showed EGFP and mCherry colocalization in cells expressing the astrocytic marker GFAP, but not in cells expressing microglial (Cd11b), or neuronal (NeuN) markers (Fig. 2c–k', Supplementary Fig. 1). In the absence of Tam induction, Aldh1l1-NuTRAP mice did not display EGFP or mCherry expression (Supplementary Fig. 2), consistent with temporally controlled, Tam-dependent induction of cre-recombinase under the control of the Aldh1l1 promoter.

**Astrocyte transcriptome enrichment in the Aldh1l1-NuTRAP mice by TRAP-RNAseq**. Enrichment of EGFP-tagged polysomes was performed with the TRAP protocol. The resulting positive and negative fractions, as well as input fraction, were collected for RNA isolation. qPCR measurements showed significant enrichment of marker genes for astrocytes (Aldh1l1, Fabp7, Gfap, Elovl2, Aqp4, and Kcnj10) in the positive fraction compared to input and negative fractions. Depletion of marker genes for microglia (Cx3cr1, C1qa, Gpr84, and Aif1), neurons (Syt2 and Syt4), and oligodendrocytes (Mog, Neu4, and Opalin) was observed in the positive TRAP fraction compared to the other fractions (Fig. 3a). RNAseq analysis, as visualized by Principal Component Analysis (PCA), revealed separation of positive fraction from input, negative, and whole tissue samples in the first component (Fig. 3b). Cell-type-specific marker gene lists were generated from prior cell-sorting studies[18] (Supplementary Data 1). The distribution of cell-type-specific gene expression showed enrichment of astrocytic genes and depletion of microglial, neuronal, and oligodendrocytic genes in the positive fraction relative to input (Fig. 3c, d).

One prior study applied the RiboTag approach with the same Aldh1l1- cre/ERT2 line of mice[12]. In another recent study[19] the RiboTag approach with a Gfap-cre was used to target the astrocyte transcriptome. We compared the lists of astrocyte marker genes (BHMTC $p < 0.05$, FC enrichment >5) generated in these studies with the data from the NuTRAP line developed here and found 127 ribosomal-tagging marker genes for astrocytes that are independent of ribosomal-tagging approach or cre line (Fig. 3e, Supplementary Data 1). Pairwise correlation of positive fraction/input ratios for all expressed genes demonstrate a high level of overall correlation even if differences in genes achieving the 'marker' ($p < 0.05$, FC > 5) sets was evident (Fig. 3f). When a list of 561 ribosomal-tagging astrocyte marker genes (the sum of genes in common between at least two of the studies in Fig. 3e) was compared to previously identified astrocyte markers from cell-sorting studies[18], we found 88 isolation method-independent astrocyte marker genes (Fig. 3g, Supplementary Data 1). Additional comparisons of enrichment distribution of gene expression between the different ribosomal profiling methods and gene marker lists from cell-sorting studies were performed (Supplementary Fig. 3). Taken together, these comparisons demonstrate a commonality to astrocyte enriched genes with some minor differences in RiboTag versus NuTRAP and Aldh1l1 versus Gfap cre lines. Astrocyte enriched transcripts further demonstrated

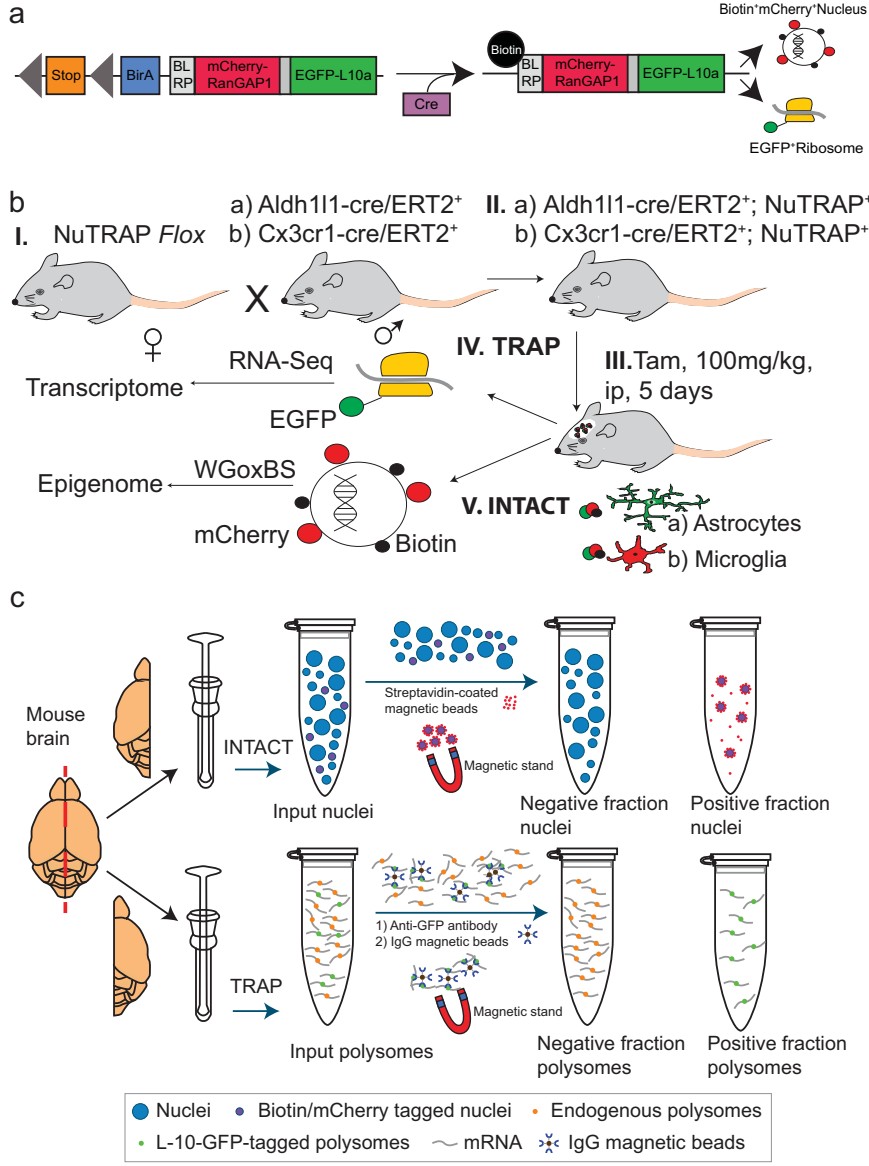

**Fig. 1 Main experimental design in the study. a** Representation of the NuTRAP construct and its recombined products upon cre-mediated induction. **b** Breeding strategy to achieve astrocyte- and microglia- specific RNA and DNA. **c** Schematic of the cell-specific RNA and DNA isolation from the same mouse brain via the TRAP and INTACT methods. Note: currently TRAP and INTACT from the same homogenate is not possible, but parallel structures from each hemisphere of the same mouse can be used for paired analyses.

over-representation of genes critical in astrocyte physiological functions[12,19,20], such as cholesterol synthesis and transport, fatty acid metabolism, receptors/channels, and synapse modification (formation, function, and elimination), while under-representation of complement/immune mediators, commonly associated with microglial function (Supplementary Fig. 4). These findings are collectively in agreement with the normal physiology of astrocytes in the brain and demonstrate specific targeting and enrichment of astrocyte transcripts in the Aldh1l1-NuTRAP model.

**Validation of astrocytic epigenome enrichment in the Aldh1l1-NuTRAP mouse brain by INTACT-BSAS.** Nuclear preparations of Aldh1l1-NuTRAP were subjected to INTACT isolation with streptavidin magnetic beads for separation of negative and positive (biotinylated) nuclei. To assess purity of putative astrocytic nuclei in the positive fraction, nuclei were evaluated for

expression of mCherry by confocal microscopy imaging (Fig. 4a, b). Biotinylated nuclei covered by streptavidin beads (fluoresce in the red channel[10]) were evident in the positive fraction (Fig. 4a) and comprised 90% of the positive fraction (Fig. 4b). With the predicate that mCG in gene promoters is inversely related to transcriptional activation, Bisulfite Amplicon Sequencing (BSAS) analysis was performed to measure the percentage genomic CG methylation (mCG) in the promoter region of selected astrocytic (Aldh1l1, Fabp7, Gfap, and Kcnj10: Fig. 4c), microglial (Cx3cr1, C1qa, Gpr84, and Aif1: Fig. 4d), and neuronal (Eno2, Syt2, and Syt4: Fig. 4e) marker genes. Hypomethylation of astrocyte marker genes and in most cases hypermethylation of microglia and neuron marker genes was observed, in general agreement with the inverse relationship of methylation to gene expression around the TSS, where BSAS was performed. Together these data support the astrocytic identity of the INTACT-isolated nuclei and DNA in the positive fraction.

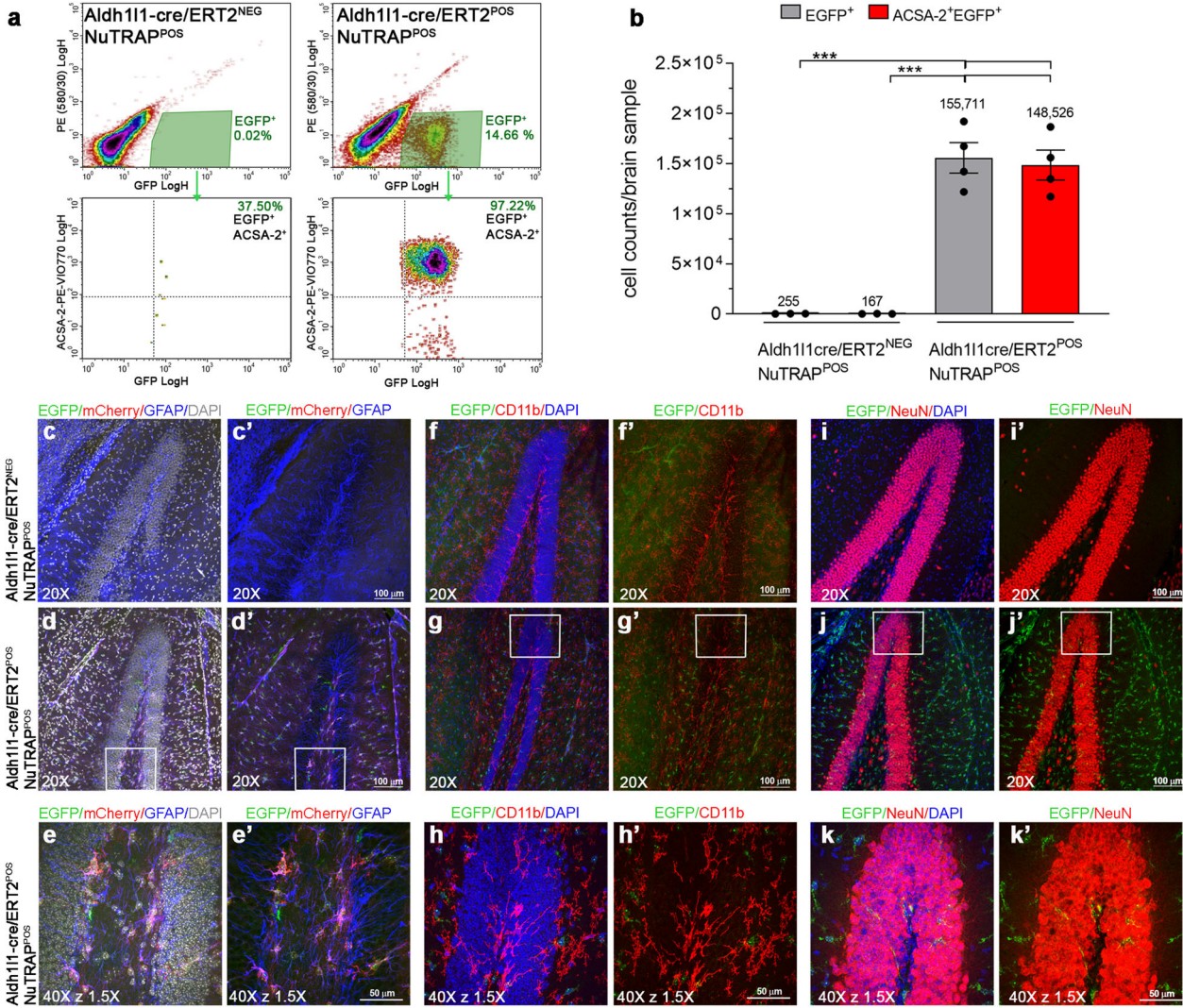

**Fig. 2 Flow cytometry and immunohistochemical validation of the Aldh1l1-NuTRAP transgene expression.** One week after Tam treatment, brains were harvested from Aldh1l1-NuTRAP and cre-negative NuTRAP+ (control) mice for flow cytometry (FC) and immunohistochemistry (IHC) purposes. **a** Representative FC plots of immunostained single-cell suspensions showed a distinct population of brain EGFP+ cells, identified as Aldh1l1+ cells (astrocyte lineage), based on gating strategy for EGFP and ACSA-2 co-expression in Aldh1l1-NuTRAP samples but not in the controls. **b** Analysis of absolute cell counts from FC quantitation expressed as mean cell count/brain sample ±SEM. **c–k'** Representative confocal fluorescent microscopy images of sagittal brain sections captured in the dentate gyrus of the hippocampus show EGFP expression (green signal) in cells that co-expressed mCherry (red signal) and GFAP (blue signal) in Aldh1l1-NuTRAP brains but did not colocalize with other cell-type marker expression. ***$p < 0.001$ between depicted groups by one-way ANOVA followed by the Tukey's multiple comparison test ($n = 4$ for cre+ group, $n = 3$ for cre− group). Scale bar at ×20 magnification: 100 μm and scale bar at ×40 z×1.5: 50 μm.

**Flow cytometry and immunohistochemical validation of the Cx3cr1-cre/ERT2+; NuTRAP+ mouse brain.** Validation of the Cx3cr1-NuTRAP line for analysis of microglia was performed with a similar approach as above. Tam was administered for 5 consecutive days and in order to avoid labeling of circulating monocytes in the tissue, which unlike resident microglia are short-lived and rapidly renew themselves[21], brain tissue collection was delayed until 3–4 weeks after treatment. Single-cell suspensions of brain tissue immunostained with antibody against CD11b, a microglia marker, revealed a distinct EGFP+ cell population present in the Cx3cr1-NuTRAP brains but not in the cre-negative subjects, consistent with the reported 5–10% microglial constituency of the mouse brain[22]. The EGFP+ cell population almost completely co-expressed CD11b, evidence of cell-specific cre recombination for the microglial lineage (Fig. 5a, b). The evidence for microglia-specific recombination was next tested with IHC. Sagittal brain sections immunostained with cell-specific markers showed EGFP and mCherry colocalization in cells expressing CD11b (Fig. 5c–h', Supplementary Fig. 5). In the absence of Tam induction, FC immunolabeling indicated that Cx3cr1-NuTRAP mice displayed a small EGFP+ cell population that mostly expressed CD11b. This agrees with reported findings using the same cre line[5], and was not clearly detected with the sensitivity of IHC (Supplementary Fig. 6).

**RNAseq validation of microglial transcriptome enrichment in the Cx3cr1-NuTRAP mouse brain by TRAP-RNAseq.** TRAP isolation was performed as described above. Initial qPCR validation of the TRAP-isolated RNA from all three fractions showed significant enrichment of marker genes for microglia (*Cx3cr1, C1qa, Gpr84,* and *Aif1*) in the positive fraction compared to input and negative fractions. Significant depletion of marker genes for astrocytes (*Aldh1l1, Gfap, Aqp4,* and *Kcnj10*), oligodendrocytes

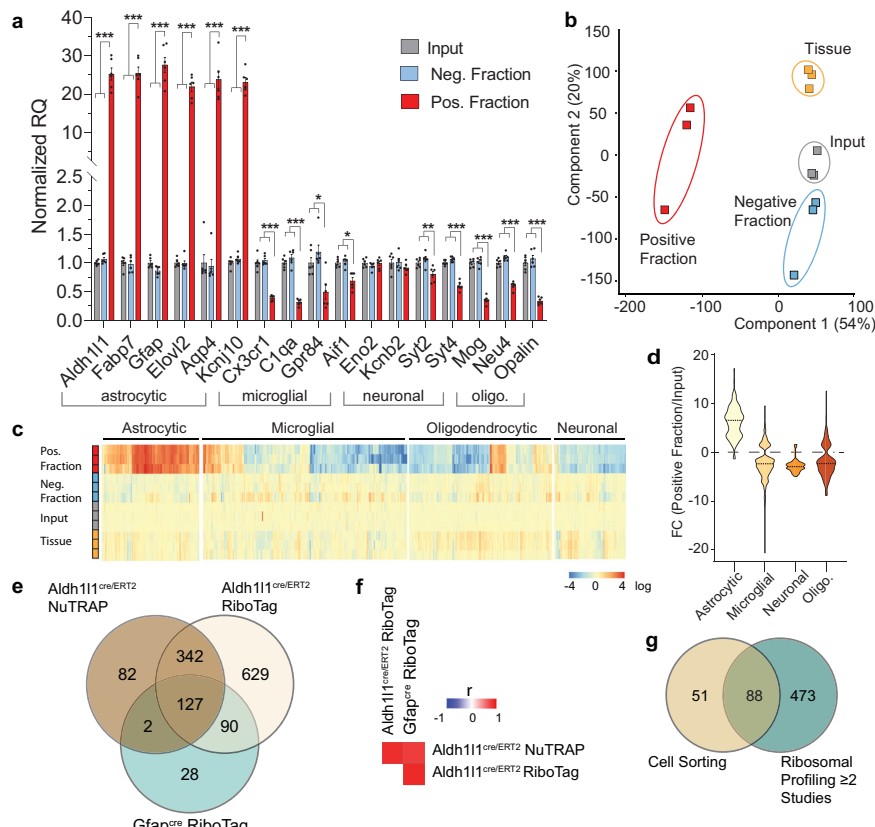

**Fig. 3 Transcriptomic validation of astrocytic enrichment in TRAP-RNA from Aldh1l1-NuTRAP mouse brain. a** TRAP-isolated RNA from input, TRAP-negative and TRAP-positive fractions were examined by qPCR for enrichment and depletion of selected cell-specific genes for astrocytes, microglia, neurons, and oligodendrocytes. Bar graphs represent mean relative gene expression ± SEM for each gene measured. *$p < 0.05$, **$p < 0.01$, ***$p < 0.001$ by RM one-way ANOVA with Benjamini–Hochberg multiple testing correction followed by Tukey's multiple comparison test across fractions ($n = 6$/group). **b** RNAseq analysis was performed on all fractions and total brain RNA ($n = 3$/group). Principal component analysis of transcriptome profiles showed separation of positive fraction from input, negative, and tissue samples by the first component. **c** Expression of cell-type marker gene lists, generated from cell-sorting studies shows enrichment of astrocytic genes and depletion of other cell-type marker genes in the positive fraction versus other fractions. **d** Enrichment or depletion of marker genes is presented as the fold change (Positive fraction/Input). Astrocyte marker genes were enriched in the positive fraction while genes from other cell types were generally depleted in the positive faction relative to input. **e** Astrocyte marker genes identified in prior Ribo-Tag analysis (FC > 5 Positive fraction/Input) with the same cre line[12] and with a Gfap-cre line[19] were compared to the marker genes identified from the Aldh1l1-NuTRAP, demonstrating 127 ribosomal-tagging common astrocyte marker genes. **f** Pearson correlation of the fold change (Positive fraction/Input) for all expressed genes observed in all ribosomal profiling studies have similar levels of transcriptome enrichment and depletion. **g** Astrocyte markers from at least two ribosomal profiling studies were compared to astrocyte markers from cell-sorting studies[18] to identify 88 isolation method-independent astrocyte markers.

(*Mog, Neu4,* and *Opalin*), as well as for neurons (*Eno2, Kcnb2, Syt2,* and *Syt4)*, was observed in the positive TRAP fraction compared to the other fractions (Fig. 6a). RNAseq was performed on input, negative, and positive fractions from TRAP isolation, as well as whole tissue. Transcriptome profiles revealed separation of positive fraction from all other groups by PCA (Fig. 6b). Fold change enrichment in the positive TRAP fraction versus the input was compared to microglial marker genes lists from cell-sorting studies. Enrichment of microglial genes and depletion of astrocytic, neuronal, and oligodendrocytic genes was observed in the positive fraction relative to input (Fig. 6c, d). The same Cx3cr1-cre/ERT2(Jung) line as used here has been used with RiboTag enrichment of microglial RNA[5]. In another study, the Cx3cr1-cre/ERT2 + (Litt) line was crossed with a TRAP mouse model[3]. We compared the lists of microglial marker genes with FC > 5 ($p < 0.05$, positive fraction/input) in these studies[3,5] with the Cx3cr1-NuTRAP (present study). We identified 142 ribosomal-tagging common microglial marker genes (Fig. 6e, Supplementary Data 2). Pairwise correlation of positive fraction/input ratios for all expressed genes demonstrate a high level of overall correlation

even if differences in 'marker' ($p < 0.05$, FC > 5) gene sets was evident (Fig. 6f). When a list of 484 ribosomal-tagging microglial marker genes (the sum of genes in common between at least two of the studies in Fig. 6e) was compared to previously identified microglial markers from cell-sorting studies[18], we found 209 isolation method-independent microglial marker genes (Fig. 6g, Supplementary Data 2).

Genes enriched in the microglia transcriptome in our study included an over-representation of genes regulated by *PU.1* (also known *as Spi1*), a transcription factor that shapes the homeostatic functions of microglia[23] (Supplementary Fig. 7). Collectively, data provide ample support that the Cx3cr1-NuTRAP model is suitable for studying the microglia transcriptomic signatures of the brain in both homeostatic and stress settings.

The successful enrichment of the microglial transcriptome in smaller tissue dissections was also performed, in this case on the hippocampus (Fig. 7a–d). Similar enrichment and depletion of marker genes and cross study correlations demonstrate that NuTRAP can be scaled down to small, dissected brain regions. Moreover, additional comparisons of enrichment distribution of

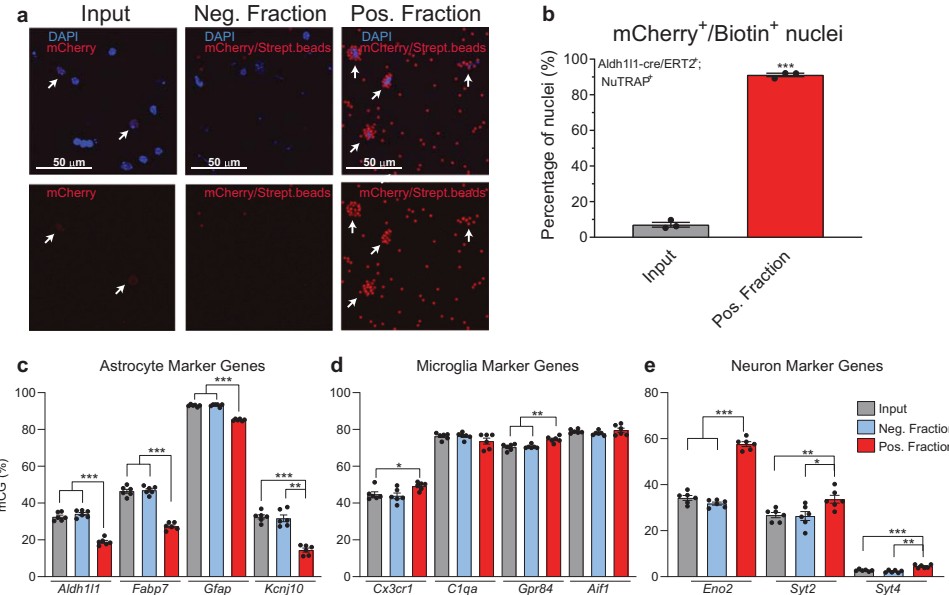

**Fig. 4 Validation of astrocytic nuclei and epigenome enrichment in the Aldh1l1-NuTRAP mouse brain by INTACT-BSAS. a** Representative confocal fluorescent microscopy images from input, negative, and positive INTACT nuclei fractions. Scale bar: 50 µm. **b** Purity of astrocytic nuclei, expressed as average percentage ± SEM mCherry+/ Biotin+ nuclei in the positive fraction, compared to percentage ± SEM mCherry+ nuclei in the input demonstrates a high degree of specificity to the INTACT isolation ($n = 3$/group, ***$p < 0.001$ by paired $t$-test comparing positive fraction to input). **c–e** INTACT-isolated genomic DNA from Aldh1l1-NuTRAP mice was bisulfite converted and DNA methylation in specific regions of interest (promoters for neuron, astrocytes and microglia marker genes) were analyzed by Bisulfite Amplicon Sequencing (BSAS) from input, negative, and positive fractions. Hypomethylation of the astrocyte marker genes *Aldh1l1, Fabp7, Gfap,* and *Kcnj10* in the positive fraction compared to input and negative fraction and hypermethylation of the microglial genes *Cx3cr1* and *Gpr84,* and hypermethylation of the neuronal marker genes *Eno2, Syt2,* and *Syt4* was observed ($n = 6$/group, average % mCG ±SEM, RM One-way ANOVA with Tukey's post-hoc, *$p < 0.05$, **$p < 0.01$, ***$p < 0.001$).

gene expression between the different microglia ribosomal profiling methods and gene marker lists from cell-sorting studies demonstrated high level of concurrence between our model and approach and that of other research groups (Fig. 7e–g, Supplementary Fig. 8).

Transcriptome comparison between Aldh1l1-NuTRAP and Cx3cr1-NuTRAP positive fractions by regulator and pathway analyses also confirmed cell-specific enrichments in agreement with brain astrocytes and microglia, respectively (Supplementary Fig. 9).

**Validation of microglial epigenome enrichment in the Cx3cr1-NuTRAP mouse brain by INTACT-BSAS.** In parallel with the TRAP protocol described above, nuclear preparations of Cx3cr1-NuTRAP were subjected to INTACT isolation with streptavidin magnetic beads for separation of negative and positive (biotinylated) nuclei. To assess purity of putatively microglial nuclei in the positive fraction, nuclei were evaluated for expression of mCherry by confocal microscopy imaging. Biotinylated nuclei were covered by streptavidin beads (Fig. 8a) and reached over 90% purity in the positive fraction (Fig. 8b). CG methylation around the promoter region of selected microglial (*Cx3cr1, C1qa, Aif1,* and *Gpr84*), genes (Fig. 8c–e) demonstrated hypomethylation of the microglial gene promoters, as compared to input and negative fractions. Some astrocyte (*Aldh1l1, Gfap, Kcnj10,* and *Fabp7*), and neuronal markers *(Eno2, Syt2,* and *Syt4)* also demonstrated hypermetylation in the positive fractions. These findings were indicative of the microglial identity of the nuclei isolated in the positive fraction by INTACT.

**Cell-specific epigenetic findings by whole-genome oxidative bisulfite sequencing comparing Aldh1l1-NuTRAP and Cx3cr1-NuTRAP models.** The landscape of the brain epigenome at a

single-base resolution, and at the cell-type-specific level, remains largely unknown[2,24]. Moreover, comparison of DNA modifications, both methylation and hydroxymethylation in CG and non-CG contexts, of different cell types, such as astrocytes and microglia, using the combination of inducible cre-recombinase and NuTRAP technologies has not been previously performed. Upon validation of the cell-specific identity of the INTACT-isolated gDNA from positive fractions by BSAS (Fig. 4 and Fig. 8), WGoxBS sequencing libraries were constructed from the DNA samples isolated from input, negative, and positive INTACT fractions. Genome-wide levels of mCG, hmCG, mCH, and hmCH (see Supplementary Fig. 10 for conversion efficiency controls) were compared across fractions and cell types. The analysis revealed that global mCG levels are similar (~70%) between the Aldh1l1-NuTRAP and Cx3cr1-NuTRAP positive fractions (Fig. 9a). Of interest, levels of hmCG were lower in the Cx3cr1-NuTRAP positive fraction as compared to input and Aldh1l1-NuTRAP positive fraction (Fig. 9b) and mCH levels were lower in both positive fractions as compared to input and negative fractions (Fig. 9c). These data demonstrate that microglia contain less cytosine hydroxymethylation compared to other cell types including astrocytes (Fig. 9b). The analysis of mCH levels showed a significantly lower level of mCH in the Aldh1l1-NuTRAP and Cx3cr1-NuTRAP positive fractions with respect to their negative fractions. The lower level of non-CG methylation was more pronounced in microglia, being significantly less than the input. This is consistent with the concept that mCH is concentrated in neurons[1] and provides more specific detail that this is true when astrocytes or microglial alone are examined. As previously reported for the brain[14,25,26], non-CG hydroxymethylation (hmCH) was not detected in any of the samples analyzed (Fig. 9d). Principal component analysis of CG methylation across astrocytic, microglial, neuronal, and oligodendrocytic gene bodies separated input, Aldh1l1-NuTRAP positive, and Cx3cr1-NuTRAP positive

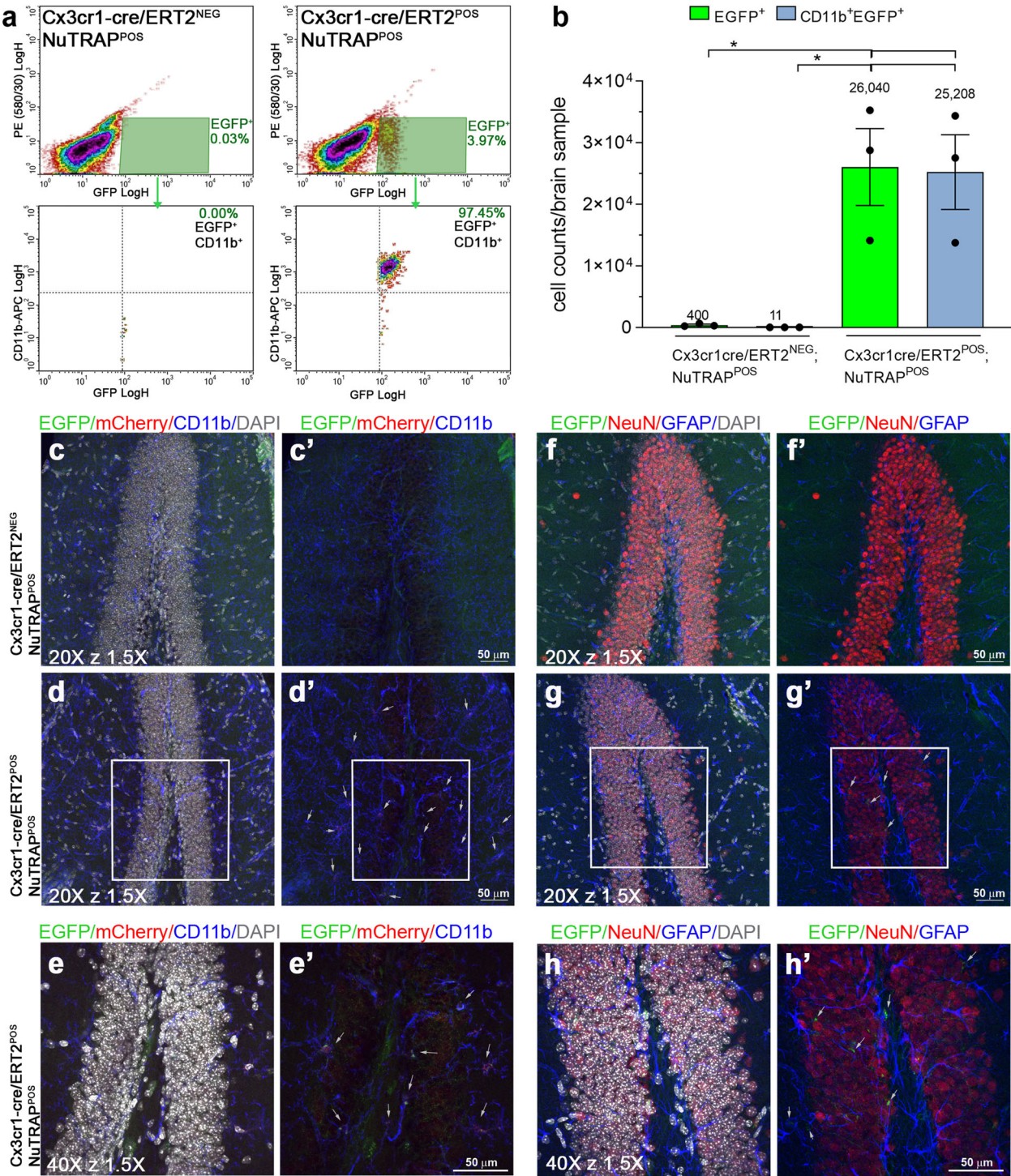

**Fig. 5 Flow cytometry and immunohistochemical validation of the Cx3cr1-NuTRAP mouse brain.** After Tam treatment, brains from Cx3cr1-NuTRAP and cre-negative NuTRAP+ (control) mice were harvested and single hemispheres assessed by flow cytometry (FC) and immunohistochemistry (IHC). **a** Representative FC plots of immunostained single-cell suspensions showed a distinct population of brain EGFP+ cells, identified as CD11b+ cells (microglia lineage), based on gating strategy for EGFP and CD11b co-expression in Cx3cr1-NuTRAP samples upon cre-mediated recombination but not in the controls. **b** Analysis of absolute cell counts from FC quantitation expressed as mean cell count/brain sample ±SEM. **c–h′** Representative confocal fluorescent microscopy images of sagittal brain sections captured in the dentate gyrus of the hippocampus. EGFP expression (green signal) was found in cells that co-expressed mCherry (red signal) and CD11b (blue signal) in Cx3cr1-NuTRAP+ brains. *$p < 0.05$ between depicted groups by one-way ANOVA followed by the Tukey's multiple comparison test ($n = 3$/group). Scale bar at ×20 z ×1.5 and scale bar at ×40 z ×1.5: 50 μm.

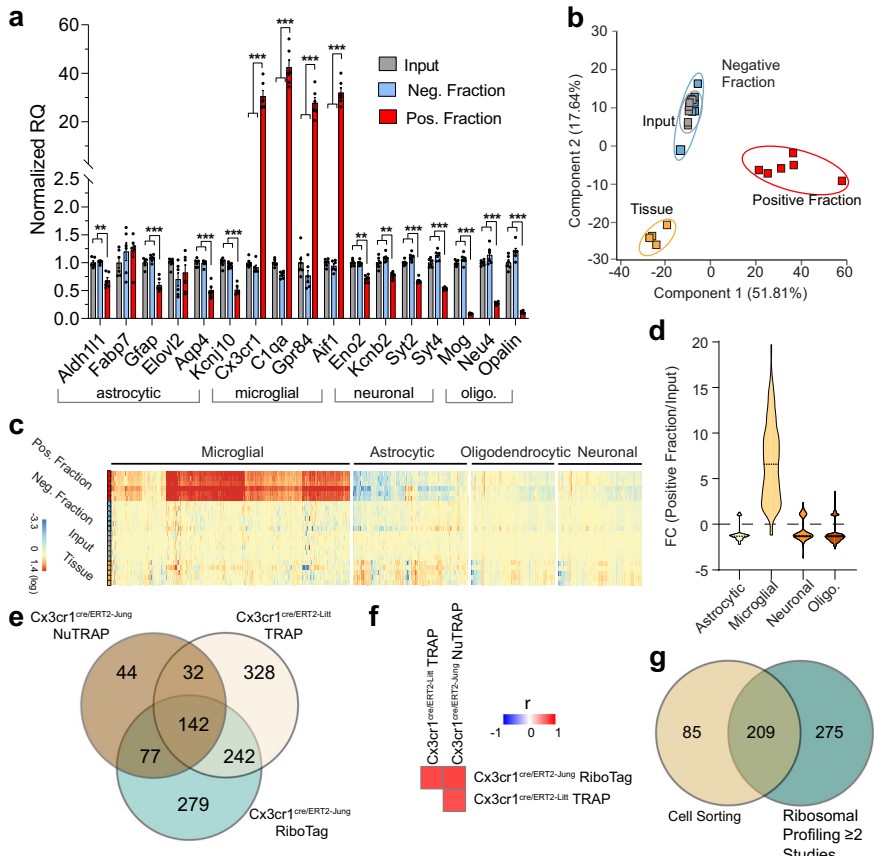

**Fig. 6 Validation of microglial TRAP-RNA enrichment in the Cx3cr1-NuTRAP mouse brain. a** TRAP-isolated RNA from input, negative, and positive fractions were examined by qPCR for enrichment/depletion of selected cell-specific genes for microglia, astrocytes, oligodendrocytes, and neurons. Bar graphs represent average relative gene expression ± SEM. *$p < 0.05$, **$p < 0.01$, ***$p < 0.001$ by RM one-way ANOVA with Benjamini–Hochberg procedure to correct for multiple comparisons of genes followed by Tukey's multiple comparison test of fractions ($n = 6$/group). **b** Principal component analysis of transcriptome profiles showed separation of positive fraction from input, negative and tissue samples by the first component. **c** RNAseq heatmap graph of cell-type marker genes from prior cell-sorting studies shows enrichment of microglial marker genes and depletion of other cell-type markers, as compared to whole tissue, input, and negative fractions. **d** Marker gene lists for different cell types were generated from cell-sorting studies as described in the text. Enrichment or depletion of genes from each of the lists is presented as the fold change (Positive fraction/Input). Microglial marker genes were enriched in the positive fraction while genes from other cell types were generally depleted in the positive fraction relative to input. **e** Microglia marker genes with FC > 5 (Positive fraction/Input) from the Cx3cr1-cre/ERT2[+] model with RiboTag[5], Cx3cr1-cre/ERT2[+] model with TRAP[3], and NuTRAP identifies 142 ribosomal-tagging common microglial marker genes. **f** Pearson correlation of the fold change (Positive fraction/Input) for all expressed genes observed in all studies have similar levels of transcriptome enrichment and depletion. **g** Comparison of 484 microglial markers from at least two ribosomal profiling studies with previously identified microglial markers from cell-sorting studies[18] identifies 209 isolation method-independent microglia marker genes.

fractions in the first and second components (Fig. 9e). To uncover potential cell-type-specific differences in mCG patterns, methylation across cell-type marker genes (from −4 kb in respect to the TSS and +4b from the TES) was compared for astrocytes (Fig. 9f) and microglia (Fig. 9i). In Aldh1l1-NuTRAP INTACT-positive fractions, but not Cx3cr1-NuTRAP INTACT-positive fractions, hypomethylation upstream, within, and downstream the gene body (Fig. 9g) was evident across astrocyte marker genes as compared to input and Cx3cr1-NuTRAP positive fractions. This correlates with the higher levels of mRNA expression of these genes in Aldh1l1-NuTRAP TRAP-positive fraction (Fig. 9h). Similarly, only in the Cx3cr1-NuTRAP INTACT-positive fraction, hypomethylation upstream, within, and downstream the gene body of microglial markers genes was evident (Fig. 9j) and in agreement with higher mRNA expression of these genes in microglia (Fig. 9k).

CG dinucleotides are found far less frequently than other dinucleotide pairs (<1% dinucleotide pairs) and are clustered together in CpG islands. Definitions for the regions around the CpG islands have been established and include shores (2Kb

upstream and downstream from CpG island) and shelves (2Kb upstream and downstream from shores). Despite their high CG content, CpG islands are mainly unmethylated while methylation is higher in shores and shelves[27]. Analysis of methylation and hydroxymethylation levels covering CpG islands, shores, and shelves revealed that the shores and shelves of Cx3cr1-NuTRAP INTACT-positive fractions had significantly higher mCG levels (Supplementary Fig. 11a, b) and significantly lower hmCG levels (Supplementary Fig. 11c, d) compared to the other groups. The findings allow us to speculate that while low levels of mCG and hmCG are conserved in CpG islands across the genome, epigenetic signatures found in shores and shelves might differentiate microglia from the other cell types of the brain.

More than two-thirds of the mammalian genome consists of repetitive elements[28], including long terminal repeats (LTR), long interspersed nuclear elements (LINE), short interspersed nuclear elements (SINE), major satellites, and simple repeats[29]. The biological significance of repetitive element methylation/hydroxymethylation is unknown and has been difficult to explore in a cell-type-specific manner. Input and positive fractions from

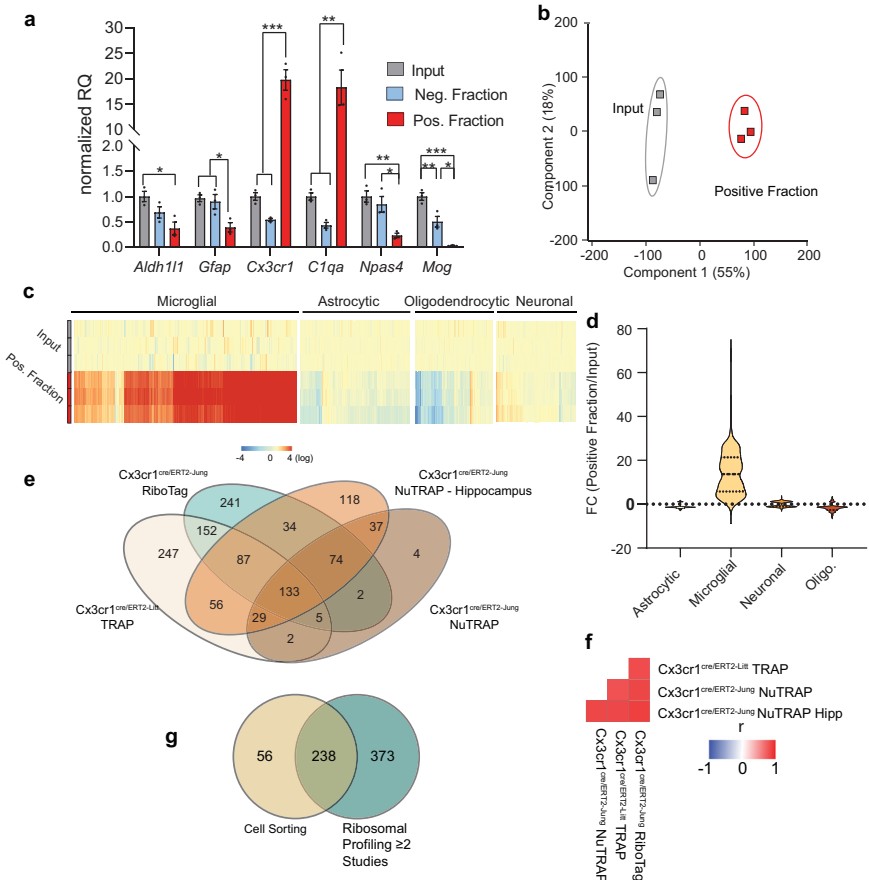

**Fig. 7 The enrichment of microglial transcriptome in the Cx3cr1-NuTRAP brain is scalable to small brain regions such as hippocampus. a** TRAP-isolated RNA from input and positive fractions of dissected hippocampus were examined by qPCR for enrichment/depletion of selected cell-specific genes for astrocytes, microglia, neurons, and oligodendrocytes. Bar graphs represent average relative gene expression ± SEM. *$p < 0.05$, **$p < 0.01$, and ***$p < 0.001$, respectively, by one-way ANOVA with Tukey's multiple comparison test of fractions ($n = 3$/group). **b** Principal component analysis of transcriptome profiles showed separation of positive fraction from input. **c** RNAseq heatmap graph of cell-type marker genes from prior cell-sorting studies shows enrichment of microglial marker genes and depletion of other cell-type markers, as compared to input. **d** Marker gene lists for different cell types were generated from cell-sorting studies as described in the text. Enrichment or depletion of genes from each of the lists is presented as the fold change (Positive fraction/Input). Microglial marker genes were enriched in the positive fraction while genes from other cell types were generally depleted in the positive fraction relative to input. **e** Microglia marker genes with FC > 5 (Positive fraction/Input) from the Cx3cr1-cre/ERT2[+] model with RiboTag[5], Cx3cr1-cre/ERT2[+] model with TRAP[3], Cx3cr1-NuTRAP hemisected brain, and Cx3cr1-NuTRAP hippocampus identifies 133 ribosomal-tagging common microglial marker genes. **f** Pearson correlation of the fold change (Positive fraction/Input) for all expressed genes observed in all studies have similar levels of transcriptome enrichment and depletion. **g** Comparison of 611 microglial markers (enriched in at least two ribosome profiling studies in **e**) with previously identified microglial markers from cell-sorting studies[18] identifies 238 isolation method-independent microglia marker genes.

Aldh1l1-NuTRAP and Cx3cr1-NuTRAP brain samples were analyzed for mCG, hmCG, and mCH content in whole genome, repeats, and non-repeats (Supplementary Fig. 12a–c). In general, there were either no or minimal differences in mCG levels evident in repeat elements. However, hmCG and mCH repeat element (SINE, LINE, LTR, and simple repeat) levels were lower in the Cx3cr1-NuTRAP INTACT-positive fraction as compared to the other groups (Supplementary Fig. 12d–f). These findings identify epigenetic markers that are microglia- and repetitive element-specific.

To investigate the inverse correlation of gene promoter methylation and gene expression (Fig. 9) at a single-gene resolution, paired, targeted BSAS and qPCR data for cell-specific markers was collected. Base-specific differences in positive fractions relative to input and between positive fractions was observed in all genes examined (Fig. 10a–h). Astrocyte marker genes demonstrated the lowest methylation levels in Aldh1l1-NuTRAP positive fraction DNA (Fig. 10a–d) while microglia marker genes demonstrated the lowest methylation

levels in Cx3cr1-NuTRAP positive fraction DNA (Fig. 10e–h). Of note is that methylation levels vary by the specific site across the regions examined but the topography was largely retained across samples, just shifted to higher or lower methylation levels.

As noted, there was lower whole-genome levels of mCH in the positive fractions of Aldh1l1-NuTRAP and Cx3cr1-NuTRAP brains (Fig. 9c), with absolute mCH levels less than 1%. While most sites within the astrocytic and microglial genes analyzed had mCH levels less than 1% across all fractions, there were selected sites within each region with consistently higher mCH ranging from 2–60% (Supplementary Fig. 13a–h). For astrocytic (Supplementary Fig. 13a–d) and microglial (Supplementary Fig. 13e–h) marker genes, there were several sites with statistically significant mCH differences between input, Cx3cr1-NuTRAP positive, and/ or Aldh1l1-NuTRAP positive fractions.

When CG methylation levels were averaged across sites within a region examined and correlated to paired TRAP-qPCR from the same mice, significant associations were observed for most genes (Supplementary Fig. 14). Comparing the significant correlations

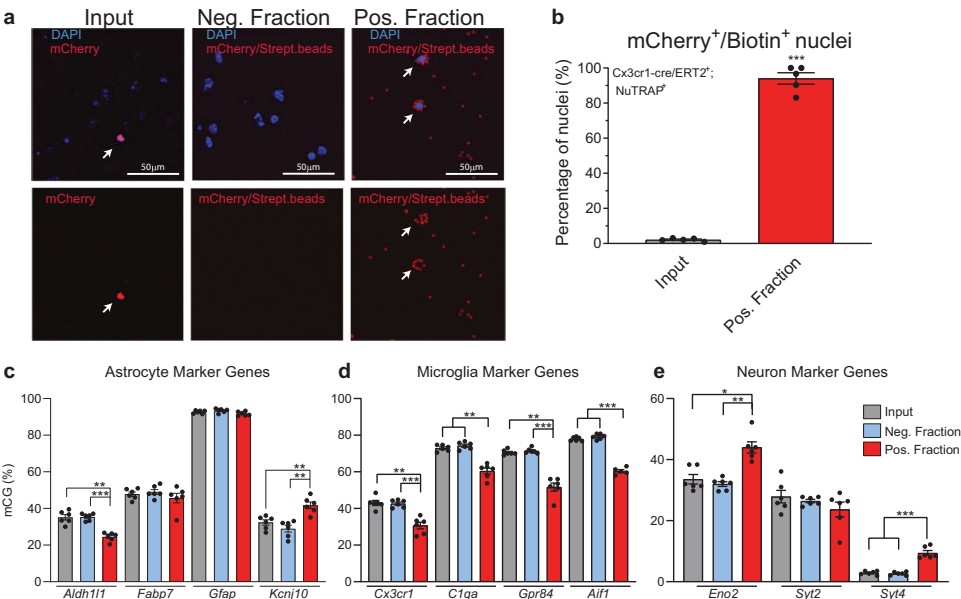

**Fig. 8 Validation of microglial epigenome enrichment in the Cx3cr1-NuTRAP mouse brain by INTACT-BSAS. a** Representative confocal fluorescent microscopy images from input, negative, and positive INTACT nuclei fractions. Scale bar: 50 μm. **b** Purity of microglial nuclei expressed as average percentage ± SEM mCherry+/ Biotin+ nuclei in the positive fraction, and percentage ± SEM mCherry+ nuclei in the input and average percentage ± SEM mCherry+/Biotin+ nuclei in the input demonstrates a high degree of specificity to the INTACT isolation ($n = 5$/group, ***$p < 0.001$ by paired $t$-test comparing positive fraction to input). **c–e** INTACT-isolated genomic DNA from Cx3cr1- NuTRAP mice was bisulfite converted and DNA methylation in specific regions of interest (promoters for neuron, astrocytes and microglia marker genes) were analyzed by Bisulfite Amplicon Sequencing (BSAS) from input, negative, and positive fractions. Hypomethylation of the microglial marker genes *Cx3cr1, C1qa, Gpr84*, and *Aif1* in the positive fraction compared to input and negative fraction was observed. Hypermethylation of neuronal markers *Eno2* and *Syt4* and astrocyte marker *Kcnj10* was observed ($n = 6$/group, average % mCG ±SEM, RM One-way ANOVA with Tukey's post-hoc, *$p < 0.05$, **$p < 0.01$, ***$p < 0.001$).

between DNA methylation and gene expression in input and positive fractions across Cx3cr1-NuTRAP and Aldh1l1-NuTRAP models reveals strong cell identity correlations. Specifically, the Aldh1l1-NuTRAP model shows strong negative correlations ($p < 0.0045$) between DNA methylation and gene expression in astrocyte marker genes (*Aldh1l1, Kcnj10, Fabp7, Gfap*). Similarly, the Cx3cr1-NuTRAP model shows strong negative correlations ($p < 0.0045$) between DNA methylation and gene expression in microglial marker genes (*Cx3cr1, Cx3cr1, Gpr84, Aif1*). Neuronal markers, in most cases showed a negative correlation in Cx3cr1- and Aldh1l1-NuTRAP mice. Interestingly, there is a strong, positive correlation between DNA methylation and qPCR for *Aldh1l1* in the Cx3cr1-NuTRAP model. The mCG levels are higher in the input and Cx3cr1-NuTRAP positive fraction as compared to Aldh1l1-NuTRAP-positive fraction. Together, these data suggest that microglia and astrocytes have cell-type-specific mCG and mCH patterning that correlates with gene expression.

**RNAseq analysis of microglial transcriptome 24 h after LPS challenge in the Cx3cr1-NuTRAP mouse brain.** To probe the utility of using NuTRAP models to identify cell-type-specific molecular changes not observable in tissue-level analyses, we performed an acute LPS administration paradigm in the Cx3cr1-NuTRAP model. Systemic delivery of LPS is commonly used to study microglial responses in the brain[30–32]. Toll-like receptors (TLRs) are pattern recognition receptors expressed by innate immune cells, such as microglia, and recognize and respond to conserved structural motifs called pathogen-associated molecular patterns (PAMPs) including LPS, initiating a cascade of molecular reactions resulting in the upregulation of proinflammatory cytokines and chemokines[33].

To interrogate the microglial transcriptome and epigenome, Cx3cr1-NuTRAP mice were administered a single i.p. injection of

5 mg/kg LPS or PBS as sham control. To confirm induction of inflammation by LPS, plasma and brain tissues were analyzed for content of inflammatory cytokines. Circulating IL-6, TNF, and IFNγ contents were elevated as early as at 4 h post LPS treatment and specifically IL-6 remained elevated in plasma and brain after 24 h (Supplementary Fig. 15a, b). Brain sections were also immunostained with mCherry and CD11b antibodies to visualize the specificity of cre-mediated recombination in microglial cells in both treatment groups (Supplementary Fig. 15c–f"). At 24 h post LPS or PBS injection, brains from Tam-induced Cx3cr1-NuTRAP mice were collected for TRAP and INTACT protocols. Initially, TRAP-isolated RNA samples were processed for qPCR analysis of genes associated with microglia and downstream activation of the TLR4 pathway in input, negative, and positive fractions (Supplementary Fig. 16). Microglial markers *C1qa* and *Itgam*, along with *Tlr4*, were highly enriched in the positive fraction of PBS- and LPS-treated mice compared to all input and negative fractions. Additionally, significantly higher induction of *Myd88, Il1α, Il1β*, and *Tnfα* was evident in LPS TRAP samples but not in LPS input or negative fractions relative PBS control treatment (Supplementary Fig. 16). In the design of the RNAseq experiment, libraries were made from RNA from input and positive TRAP fractions, excluding the negative fraction for further analyses. PCA revealed separation of samples by fractionation in the first component, and separation of samples by treatment in the second component (Fig. 11a). Differential gene expression in response to LPS was compared between positive fraction and input. LPS-induced changes demonstrate higher fold changes when microglial RNA is isolated by TRAP (Fig. 11b) as also evident in heatmap presentation with hierarchical clustering of gene expression that differentiated input and positive fractions first and secondly by treatment (Fig. 11c). Collectively, the data suggest that the NuTRAP approach produced excellent microglia-specific gene enrichment,

and microglial responses to a stimulus, such as LPS, can be revealed, or are more pronounced when compared to analysis of whole tissue.

Although a handful of studies have suggested DNA methylation as a principal regulator of microglial activation[34], little microglia-specific in vivo evidence is available to compare DNA methylation with concurrent changes in transcriptomic response.

By coupling LPS administration with the cell-type-specific Cx3cr1-NuTRAP model we are able to interrogate dynamic changes in DNA methylation in Cx3cr1[+] (microglia) cells with their paired transcriptomic changes indicative of a proinflammatory response. Cx3cr1-NuTRAP mice were systemically administered 5 mg/kg LPS or PBS by i.p. injection and 24 h after treatment and in parallel with the TRAP procedure, half brains

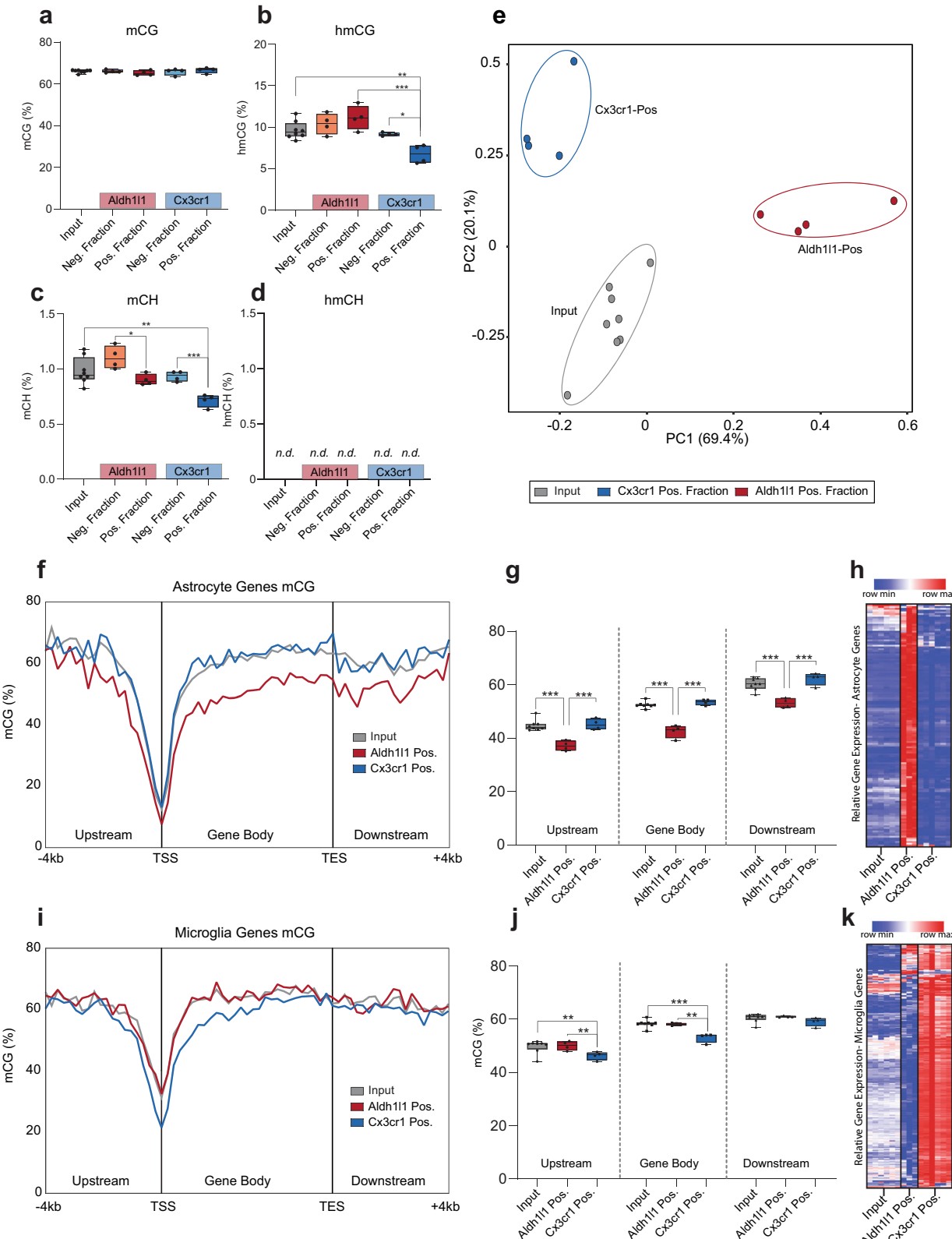

**Fig. 9 DNA modification profiles of INTACT-isolated DNA from Aldh1l1-NuTRAP and Cx3cr1-NuTRAP mouse brains by WGoxBS.** INTACT-DNA samples from Aldh1l1-NuTRAP and Cx3cr1-NuTRAP brains were used for epigenome analyses. **a–d** Total genomic levels of mCG, hmCG, mCH, and hmCH ($n = 8$/input, $n = 4$/positive fraction; One-way ANOVA with Tukey's multiple comparisons test, *$p < 0.05$, **$p < 0.01$, ***$p < 0.001$). **e** Principal component analysis of average gene body mCG (%) across astrocytic, microglial, neuronal, and oligodendrocytic cell marker lists separates input, Aldh1l1-NuTRAP-positive (Aldh1l1-Pos), and Cx3cr1-NuTRAP-positive (Cx3cr1-Pos) fractions. **f** mCG averaged over 200 nucleotide bins upstream, in the gene body, and downstream of published astrocyte genes (McKenzie)[18] in the positive fraction of Aldh1l1-NuTRAP, positive fraction of Cx3cr1-NuTRAP, and input samples combined. **g** Average percentage mCG in the positive fraction of Aldh1l1-NuTRAP, positive fraction of Cx3cr1-NuTRAP, and input samples combined in genomic DNA upstream 4 kb of the TSS, in the gene body, and downstream 4 kb of the TES of astrocytic genes. **h** Hypomethylation of astrocytic gene promoters in the Aldh1l1-NuTRAP-positive fraction correlates with higher astrocytic gene expression in the Aldh1l1-positive fraction than input and Cx3cr1-NuTRAP-positive fraction. **i** mCG averaged over 200 nucleotide bins upstream, in the gene body, and downstream of published microglia genes[18] in the positive fraction of Aldh1l1-NuTRAP, positive fraction of Cx3cr1-NuTRAP, and input. **j** Average percentage mCG in the positive fraction of Aldh1l1-NuTRAP, positive fraction of Cx3cr1-NuTRAP, and input DNA upstream 4 kb of the TSS, in the gene body, and downstream 4 kb of the TES of microglia genes. **k** Hypomethylation of microglia gene promoters in the Cx3cr1 positive fraction correlates with higher microglia gene expression. **e–i** $n = 8$/input, $n = 4$/positive fraction; 2-way ANOVA with Sidak's multiple comparison test, *$p < 0.05$, **$p < 0.01$, ***$p < 0.001$.

were dissected for INTACT protocol and downstream applications. BSAS analysis of selected microglial (*Gpr84*, *Aif1*), astrocytic (*Fabp7*), and neuronal (*Eno2*) marker genes was conducted on input and INTACT-isolated positive fractions. The INTACT-isolated positive fraction exhibited lower CG methylation (mCG) in the promoter region of microglial marker genes *Gpr84* and *Aif1* as compared to input, regardless of treatment (Supplementary Fig. 17a, b). Hypomethylation of the *Gpr84* and *Aif1* promoters in the positive fraction correlated with their respective increased gene expression by TRAP-RNAseq (Supplementary Fig. 17c, d). There was no correlation between gene expression and methylation for the astrocytic marker *Fabp7* or the neuronal marker *Eno2* (Supplementary Fig. 17e–h).

As the resident macrophages of the CNS, microglia are equipped with a number of TLRs, including TLR2 and TLR4. TLRs 2 and 4 recognize LPS as a PAMP and initiate an inflammatory cascade that acts through downstream mediators, like Myd88 and Ly96. To assess the effects of LPS administration on DNA methylation, BSAS of inflammatory genes (*Tlr2*, *Myd88*, *Ly96*) was conducted on INTACT-isolated DNA in parallel with TRAP-RNAseq. Upon LPS administration, the *Tlr2* promoter was hypomethylated in the positive fraction but not input, when compared to their respective vehicle controls (Fig. 11d). Hypomethylation of the *Tlr2* promoter in the positive fraction with LPS correlated with increased gene expression in the positive fraction with LPS (Fig. 11d). *Myd88*, a downstream effector of TLRs, showed decreased CG methylation in the positive fraction with LPS treatment when compared to PBS control, while the input had no change in *Myd88* methylation with LPS treatment (Fig. 11e). The change in promoter mCG in the positive fraction with LPS was correlated to an increase in *Myd88* gene expression (Fig. 11e). While the positive fraction showed a decrease in *Ly96* methylation with LPS administration (Fig. 11f), there was a trend toward increased *Ly96* gene expression (Fig. 11f). Of note, in the cases of *Tlr2*, *Myd88*, and *Ly96* promoter methylation, the changes in methylation observed in the positive fraction were not apparent in the input. This highlights the importance of studying DNA modifications in a cell-type-specific manner and the value of the Cx3cr1-NuTRAP model to study the relationship between microglia genomic methylation and transcriptome.

Lastly, to further demonstrate the utility of the NuTRAP system for additional molecular analyses, we examined microglial proliferation by stable isotopic labeling. This approach uses deuterium oxide ($D_2O$) in drinking water, which quickly equilibrates its labeling with the deoxyribose moiety[35]. The labeled deoxyribose moiety is then incorporated into DNA through de novo synthesis only, with no contribution of salvage pathways or repair processes. After 30 days of $D_2O$ administration to Cx3cr1-NuTRAP mice INTACT isolation was performed

and the DNA extracted. Incorporation of deuterium was determined through GC-MS in the positive fraction and input and found to be significantly greater in the positive fraction (Supplementary Fig. 18) indicating that microglial replication is greater than the average of all CNS cellular populations[36].

In summary, the results offer extensive evidence to support the combination of inducible cre/lox and NuTRAP models as a suitable and powerful approach for the parallel study of the cell-specific epigenetic and transcriptomic signatures in the brain.

## Discussion

Two of the main challenges that obstruct the interpretation of neuroepigenetic studies are the isolation of specific cell types from the complex milieu of the CNS and the lack of approaches to analyze both the transcriptome and epigenome of such cells. Combining TRAP and INTACT tagging into one construct that can be temporally controlled provides a tractable approach for cell-type-specific paired analysis of the epigenome and transcriptome. We present the development, validation and application of this approach for astrocytes and microglia. These approaches could be applied to any CNS cell type for which there is an appropriate cre driver line. The inducible nature of the Cre-Lox systems used (Aldh1l1-cre/ERT2[12] and Cx3cr1-cre/ERT2[13]), in combination with the recently developed NuTRAP construct[11], also allows for temporal control of labeling of cell-specific nuclei and polyribosomes, avoiding the deleterious effects of constitutive DNA recombination during development and potential confounds from having developmental expression of the cre when studying adult/aged stages of the lifespan. To the best of our knowledge, this is the first study applying the NuTRAP model to neuroscience research. Importantly, these results also provide approaches for generation and validation of NuTRAP neuroscience models crossed to any relevant cell-type-specific cre line.

The NuTRAP system combining TRAP and INTACT tagging approaches into one floxed construct was first described and applied to adipocytes[11]. The potential use in neuroscience research is relatively obvious as a number of reports describe the limitations of using cell sorting through surface markers. Importantly for glial research the very act of flow cytometry may change the activational state of these cells[5]. TRAP and INTACT isolations allow nucleic acids to be rapidly isolated from sub-cellular fractions decreasing the likelihood of isolation artifacts. Nonetheless, future applications of these mouse models include using the EGFP and/or mCherry labels to isolate specific cell types (or potentially nuclei) prior to single-cell transcriptomics or epigenomics. Focusing on a single-cell type for these analyses could provide greater insight into cell-type heterogeneity, especially cells such as these glial populations which represent a

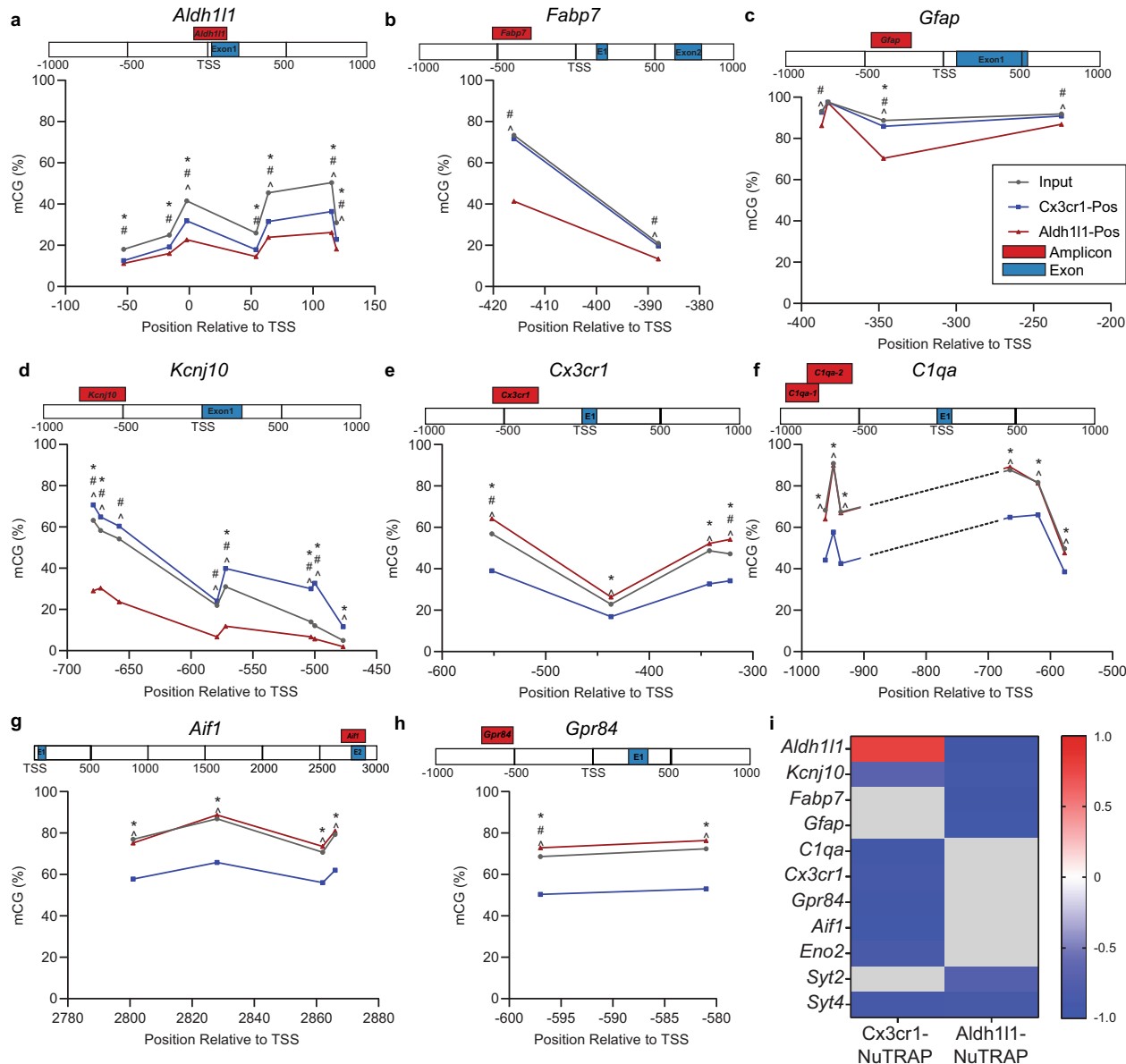

**Fig. 10 CG methylation (mCG) in specific gene promoters and intragenic regions in Cx3cr1-NuTRAP and Aldh1l1-NuTRAP mouse brain and correlation of gene promoter methylation with gene expression. a–h** Schematic of locations of targeted BSAS amplicons relative to TSS and exons of astrocytic (*Aldh1l1, Fabp7, Gfap,* and *Kcnj10*) and microglial (*Cx3cr1, C1qa, Aif1, Gpr84*) cell marker genes. Average methylation (% mCG) at each CG site within the displayed amplicon is plotted for Input, Cx3cr1-NuTRAP positive fraction (Cx3cr1-Pos), and Aldh1l1-NuTRAP-positive fraction (Aldh1l1-Pos). Topography of site-specific mCG appears to be well-conserved across fractions, despite significant differences in overall mCG between fractions. Sites with differential methylation are noted (*n* = 6/group; Two-way ANOVA with Tukey's post-hoc; *$p < 0.05$ Input vs Cx3cr1-Pos, #$p < 0.05$ Input vs Aldh1l1, ^$p < 0.05$ Cx3cr1-Pos vs Aldh1l1-Pos). **a–d** With few exceptions, each CG site within examined astrocytic marker gene regions (*Aldh1l1, Fabp7, Gfap,* and *Kcnj10*) has lower mCG in Aldh1l1-Pos than both Input and Cx3cr1-Pos. **e–h** Most CG sites within examined microglial marker gene regions (*Cx3cr1, C1qa, Aif1, Gpr84*) are lower in Cx3cr1-Pos than both Input and Aldh1l1-Pos. **i** Average mCG across each region analyzed was correlated to gene expression (normalized RQ) (Supplemental Fig. 16). Significant correlation coefficients (Pearson r; Bonferroni correction for multiple comparisons; $p < 0.0045$) displayed in the heatmap show strong negative correlation between mCG and gene expression in: astrocytic marker genes (*Aldh1l1, Fabp7, Gfap,* and *Kcnj10*) for the Aldh1l1-NuTRAP model and microglial marker genes (*Cx3cr1, C1qa, Aif1, Gpr84*) for the Cx3cr1-NuTRAP model. Neuronal genes (*Eno2, Syt2, Syt4*) CG methylation had negative correlations to gene expression in at least one model.

relatively small fraction of the cells in the brain. The use of validated cre/ERT2; NuTRAP technology not only to tag cell-specific nuclei and polysomes but also to delete genes of interest in the cells being tagged, is a promising and yet unexplored direction to be applied in mechanistic neuroscience/cell biology studies.

Validation of inducible cell-type-specific NuTRAP models requires multiple steps to confirm the specificity of both the

NuTRAP induction and the TRAP and INTACT isolations. The flow cytometry and imaging validation experiments demonstrated Tam-dependent cell-type-specific induction of the NuTRAP construct in astrocytes and microglia. Transcriptomic studies demonstrated TRAP isolation of highly enriched astroglial RNA in the Aldh1l1-NuTRAP and microglial RNA in the Cx3cr1-NuTRAP-positive fractions isolated by TRAP procedures. Cell-type enriched genes also demonstrated commonalities with other

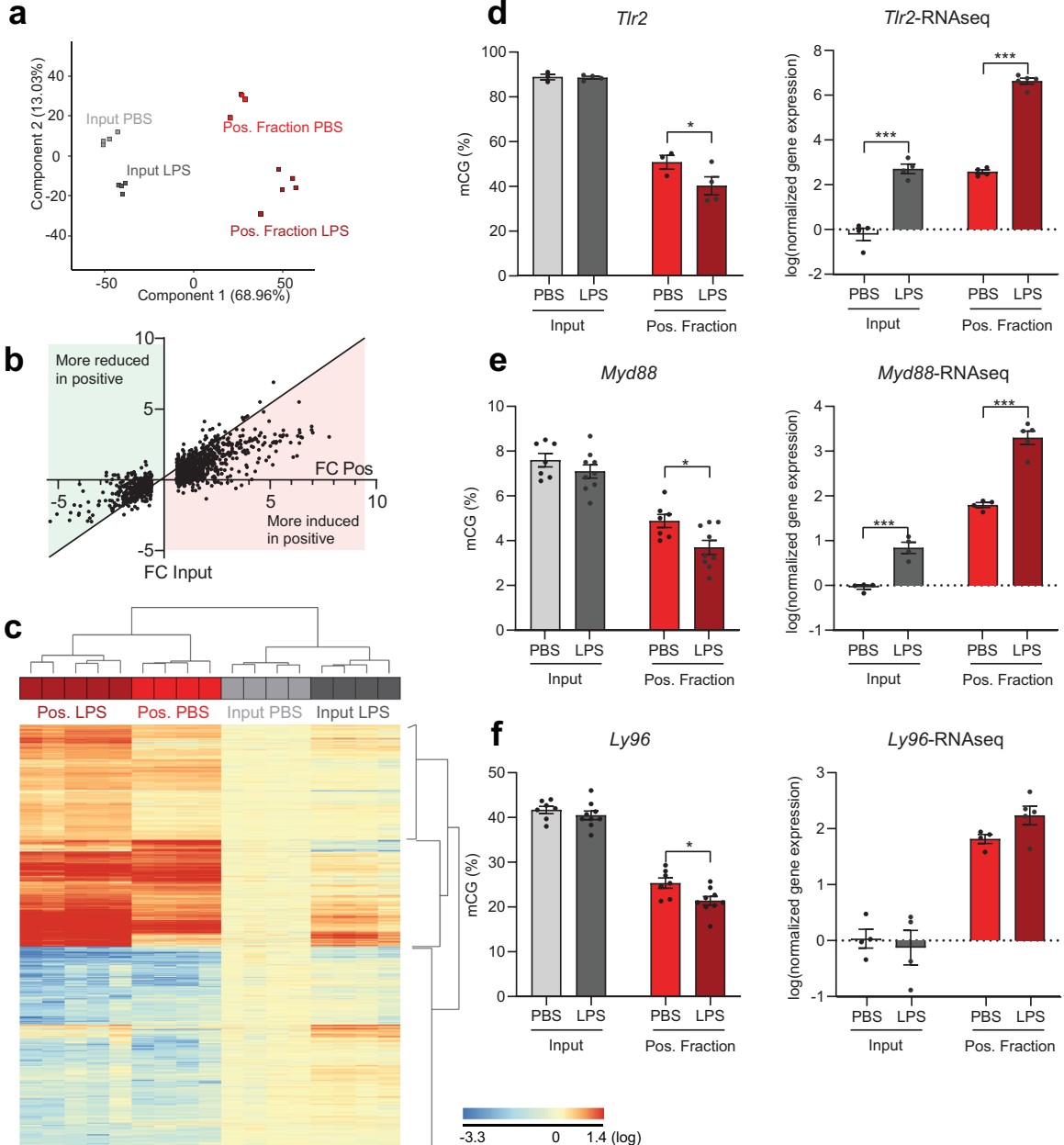

**Fig. 11 RNAseq analysis of microglial transcriptome and targeted BSAS in specific gene promoters 24 h after LPS challenge in Cx3cr1-NuTRAP mouse brain.** Cx3cr1-NuTRAP mice were treated with LPS or PBS as control for 24 h. **a** RNAseq was performed and principal component analysis of transcriptome profiles showed separation of positive fraction (PBS- and LPS-treated) from input (PBS- and LPS-treated) samples by the first component, as well as subclustering based on treatment within input and positive fraction samples by the second component. **b** Fold change of genes differentially expressed after LPS in the positive fraction were compared to the fold change in the positive fraction. LPS induced larger changes when microglial RNA is isolated by TRAP. **c** RNAseq heatmap graph of cell-type marker genes from prior cell-sorting studies shows hierarchical clustering differentiating input from positive fractions and secondly comparing treatment within type of fraction. **d, e** *Tlr2* and *Myd88* promoter methylation (mCG) decreases with LPS challenge in the positive fraction but not in the input, in correlation with increased *Tlr2* and *Myd88* gene expression in the positive fraction, as shown by RNAseq analysis. **f** *Ly96* promoter methylation decreases in the positive fraction with LPS challenge, while a trend toward increased gene expression after LPS in the positive fraction is observed by RNAseq. $n = 4$/group for *Tlr2* and $n = 7$/group for *Ly96* and *Myd88*. RNAseq: $n = 4$/PBS groups, $n = 4$/LPS input group, and $n = 5$/LPS-positive fraction group. *$p < 0.05$, **$p < 0.01$, ***$p < 0.001$ by Multiple *t*-test with Holm–Sidak correction for multiple comparisons.

ribosomal profiling studies and cell-sorting studies. Differences in the total number of enriched genes in common likely represent differences in the tissue dissected, cre lines, technical variables and the use of stringent cutoffs. When examined irrespective of cutoffs for enrichment, high levels of concordance in gene enriched was evident. Validation of cell-type-specific DNA poses a more challenging question but cell-type-specific DNA modification patterns are consistent with the concept of cell-type-

specific hypomethylation of cell marker genes. Together these studies provide high confidence that these are valid models to be applied in a broad spectrum of neuroscience research studies ranging from brain aging, to neurodegenerative and neuropsychiatric studies. Moreover, the validation approach can be applied to any new NuTRAP model and the protocols described here can be scaled-down to microdissected CNS tissue, such as hippocampus.

These findings also reveal new insights into astrocyte and microglia biology. Non-neuronal cells have been reported to have less CG hydroxymethylation than other cells in the brain and also have lower non-CG methylation[1]. While it is sometimes simplistically believed hydroxymethylation and non-CG methylation are restricted to neurons these findings provide evidence that while CG hydroxymethylation and non-CG methylation levels are lower in non-neuronal CNS cell populations they are not absent. To date, there is no explanation for why hydroxymethylation is not evident in the non-CG context but as previously reported this modification is absent or at a level below detection[14,25]. Both the genome wide and locus-specific analyses demonstrate consistent patterns of CpG island DNA modifications, lowest in the island itself with rising levels in shores and plateaus in shelves. As well, modifications presented the classic nadir near the TSS across cell types. This suggests the regulation of these patterns is common across cell types with the overall level of modification controlled in a cell-type-specific manner. This is also evident in the locus-specific analyses where the base-specific pattern of methylation was consistent across cell types and input, with the difference being in the level of modifications. The consistent topography across cell types warrants further examination as this would argue for a yet undescribed form of regulation that is independent of gene expression. This is exemplified by the BSAS analysis of cell marker genes, which differ in CG methylation level but not overall pattern. This was also observed with CH methylation. These findings allow us to speculate that CH methylation is not random and that base-specific regulation of mCH occurs along with mCG. More broadly, in relation to gene expression, previously described general principles of an inverse relationship between gene expression and promoter methylation held true in the specific cell types. Future high depth WGoxBS analysis enabled by NuTRAP models will aid the field in understanding the commonalities and differences in both the effects and regulation of DNA modifications across CNS cell types.

To determine the sensitivity of cre/ERT2-NuTRAP approaches for the detection of molecular changes at the cell-type-specific level that are not evident in tissue homogenates we acutely administered LPS to Cx3cr1-NuTRAP mice. Microglia-specific transcriptome and epigenome changes were revealed that could not be detected without affinity purification. Further, we demonstrated how our approach in combination with other labeling approaches, in this case D$_2$O, could help provide insight into how cell-specific genomic changes influence dynamic processes of the cell, such as replication. Collectively, our experiments demonstrated that the NuTRAP approach can be applied to CNS cell populations and INTACT approaches can be used to study DNA modifications, not only at the whole genome and gene promoter levels, but also in repeat elements of the genome, as shown here for the first time.

In light of the increasing interest in cell-specific contributions to the CNS epigenome[2,24] and transcriptome[3,5,12,19,20,30,31,37] landscapes, the use of transgenic inducible cre mouse models that allow for manipulation of specific floxed genes, or tagging of cell-specific nuclei/and or polysomes represent valuable research tools. Models using constructs such as INTACT[2] and RiboTag[9] constitute critical advancements for DNA or RNA studies of specific cell types. However, the introduction of inducible-cre mouse lines in combination with NuTRAP technology, as validated in this study, is a powerful strategy in the interrogation of the cell-type-specific dependent differences in the transcriptomes and epigenomes in the adult CNS.

## Methods

**Animals**. All animal procedures were approved by the Institutional Animal Care and Use Committee at the University of Oklahoma Health Sciences Center (OUHSC) and the Oklahoma Medical Research Foundation (OMRF). Mice were purchased from the Jackson Laboratory (Bar Harbor, ME), bred, and housed in the animal facility at the OUHSC and OMRF, under SPF conditions in a HEPA barrier environment. In separate breeding strategies Aldh1l1-Cre/ERT2$^{+/wt}$ males (stock number # 29655)[12] and Cx3cr1-Cre/ERT2$^{+/+}$ males (stock # 20940)[13] were mated with NuTRAP$^{flox/flox}$ females (stock # 029899)[11] to generate the desired progeny, Aldh1l1-cre/ERT2$^{+/wt}$; NuTRAP$^{flox/wt}$ (Aldh1l1-cre/ERT2$^+$; NuTRAP$^+$) and Cx3cr1-cre/ERT2$^{+/wt}$; NuTRAP$^{flox/wt}$ (Cx3cr1-cre/ERT2$^+$; NuTRAP$^+$). DNA was extracted from mouse ear punch samples for genotyping. Mice (males and females) were ~3 months old at the time of performing experiments. Euthanasia prior to tissue harvesting was carried out either by cervical dislocation, or cardiac perfusion with phosphate buffered saline (PBS), upon deeply anesthetizing mice with ketamine/xylazine. The primers used for genotyping (Integrated DNA Technologies, Coralville, IA) are included in Supplementary data 3.

**Tamoxifen (Tam) treatment**. At ~3 months of age, mice received a daily intraperitoneal (ip) injection of tamoxifen (Tam) solubilized in 100% sunflower seed oil by sonication (100 mg/kg body weight, 20 mg/ml stock solution, #T5648; Millipore Sigma, St. Louis, MO) for five consecutive days[12,14].

**Lipopolysaccharide (LPS) treatment, protein sample preparation, and suspension array**. At 3–4 weeks post-Tam treatment, Cx3cr1-cre/ERT2$^+$; NuTRAP$^+$ mice were systemically administered 5 mg/kg LPS[31] (#L2262, 1 mg/ml stock solution; Millipore Sigma) or vehicle (PBS) by ip injection. Blood was collected from the facial vein of mice at 4 h and 24 h post-LPS treatment, using a 5-mm sterile Goldenrod animal lancet (MEDIpoint, Mineaola, NY), mixed with 5 μl 0.5 M EDTA to prevent coagulation[38], and centrifuged at $1000 \times g$ for 10 min for plasma collection. At 24 h post LPS treatment, mice were euthanized and a sagittal slice circumscribing the medial line of their brains was harvested and homogenized in RIPA buffer supplemented with 1X Halt$^{TM}$ protease inhibitor cocktail (#78437; ThermoFisher Scientific, Grand Island, NY) by sonication. The supernatants from tissue homogenates were assayed for protein content using a BCA protein method (#23225; ThermoFisher Scientific) and along with diluted plasma samples, used for protein analyses. Suspension array analyte concentrations were determined using a Bio-Rad Bio-Plex System Luminex 100 and Bio-Plex manager 5.0 (Bio-Rad Laboratories, Hercules, CA)[39]. Milliplex Map luminex-based assays were used to quantify the mouse inflammatory cytokines IL-6, TNFα, and INFγ (#MCYTO-MAG-70K; EMD Millipore, Billerica, MA). The concentration of each analyte detected in plasma was expressed as log transformed (pg analyte/ml) and that detected in tissue homogenate as pg analyte/mg protein.

**Flow cytometry**. Halves of mouse brains were rinsed in D-PBS, sliced into 8–12 sagittal sections and placed into gentleMacs C-tubes, and processed for generation of single-cell suspensions using the Adult brain dissociation kit and gentleMacs$^{TM}$ Octodissociator system (#130–107–677 and #130–095–937, respectively, Milteny Biotech, San Diego, CA). The single-cell suspensions were then immunostained for flow cytometric analysis of EGFP$^+$ cell populations in the brain. The gating strategy of single cells was set to EGFP$^+$/ACSA2$^+$ for astrocytes (Supplementary Fig. 19) and EGFP$^+$/CD11b$^+$ for microglia (Supplementary Fig. 20). A 488 nm (blue) laser with 530/30 and 580/30 filter combinations was used to gate on EGFP$^+$ cells within single cells (singlets) without auto-fluorescence interference. Subsequent gating based on CD11b or ACSA-2 expression was done with 640 nm laser and 676/629 filter, or with 488 nm laser and 740 LP filter combinations, respectively. The antibodies used were anti-mouse CD11b: APC (#17–0112, clone M1/70) (eBioscience, San Diego, CA), and ACSA-2: PE-Vio770 (#130–116–246, Milteny Biotec)[15]. Isotype controls for each antibody and unstained cells were used for proper post-color compensation (Supplementary Figs. 19, 20). Samples were analyzed using a Stratedigm S1400Exi flow cytometer platform (Stratedigm, San Jose, CA) and CellCapTure v5.0 RC12 and v4.1 RC10 software (Stratedigm) at the Laboratory for Molecular Biology and Cytometry Research core facility at OUHSC. Additional gating strategies used to analyze total singlets that are EGFP$^+$ ACSA-2$^+$ (Supplementary Fig. 21) or EGFP$^+$ CD11b$^+$ (Supplementary Fig. 22) gated EGFP and glial marker expression directly on the singlet population and were used for percentage calculations (Supplementary Fig. 23).

**Immunochemistry and imaging**. For immunohistochemistry (IHC), mouse brains were harvested and hemisected. Vibratome sections: Samples were fixed for a duration of 4 h in 4% paraformaldehyde (PFA), embedded in 2% agarose, and vibratome-sectioned (Vibratome 3000 Sectioning System, The Vibratome Company, St. Louis, MO)[40]. Two-hundred-micrometer thick sagittal sections were permeated for 2 h in PBS containing 3% BSA and 0.2% Triton, and processed for fluorescence immunostaining. Frozen sections: Samples were fixed for a duration of 4 h in 4% PFA, cryoprotected by sequential incubations in PBS containing 15 and 30% sucrose, and then frozen in Optimal Cutting Temperature medium (Tissue-Tek, #4583). Twelve-micrometer thick sagittal sections were cryotome-cut (Cryostar NX70, ThermoFisher Scientific). Tissue sections were rinsed with PBS containing 1% Triton X-100, blocked for 1 h in PBS containing 10% normal donkey serum, and processed for fluorescence immunostaining. The primary

antibodies included rabbit anti-mCherry (#ab167453, 1:500, Abcam, Cambridge, MA), chicken anti-mCherry (#ab205402, 1:500, Abcam), rabbit anti-GFP (#ab290, 1:100, Abcam), chicken anti-GFAP (#ab4674, 1:1,000, Abcam), rabbit anti-NeuN (#ab177487, 1:200, Abcam), and rat anti-CD11b (#C227, 1:200, Leinco Technologies, St. Louis, MO). For confocal imaging of nuclei suspensions, unfixed, freshly isolated nuclei were mixed with DAPI solution. Sequential imaging of brain samples and freshly isolated nuclei was performed on an Olympus FluoView confocal laser-scanning microscope (FV1200; Olympus; Center Valley, PA) at the Dean McGee Eye Institute imaging core facility at OUHSC. Microscope and FLUOVIEW FV1000 Ver. 1.2.6.0 software (Olympus) settings were identical/similar for samples within experiments at same magnification. The experimental format files were.oif (4-channel capture) or.oib (2 or 3-channel capture). For brain samples, the final Z-stack generated was achieved at 1.14–1.26 µm step size with a total of 12–16 optical slices at 20X magnification (1×, 1.5×, or 2× zoom) and/or 0.55–0.62 µm step size with a total of 23–26 optical slices at ×40 magnifications (1.5× zoom). For nuclei samples, the Z-stack was achieved at 1.16 µm step size with 6–8 optical slices at ×20 magnification (2× zoom). A Zeiss Axiobserver Z1 Fluorescence Motorized Microscope (Carl Zeiss Microscopy, LLC, White Plains, NY) was used to capture entire sagittal brain sections (via Zen Blue ver.3.1 image tiling). Tiling was performed using a ×20 Plan Apo (N.A. 0.8) and an AxioCam MRm (rev 3) camera at the OMRF Imaging Core Facility. The image tiles were stitched in their native format (.czi) and the resulting composite image was compressed and exported as a.tiff file. Instrument settings for capture of raw images, as well as downstream processing (Adobe Photoshop CS5.1) of each raw image used for figure assembly are disclosed under the Supplementary data 6: equipment and settings.

**Isolation of nuclei from tagged specific nuclei (INTACT) and gDNA extraction.** The purification of viable, cell-specific nuclei from brain tissue from Tam-induced Aldh1l1-cre/ERT2[+]; NuTRAP[+] and Cx3cr1-cre/ERT2[+]; NuTRAP[+] mice was achieved by combining two previously published protocols, with modifications[10,41]. For each mouse, a hemisected half-brain was rinsed in ice-cold 1× PBS, minced into small pieces and homogenized in 4 ml ice-cold nuclei EZ lysis buffer (#NUC-101, Millipore Sigma) supplemented with 1× Halt[TM] protease inhibitor cocktail (ThermoFisher Scientific) using a glass dounce tissue grinder set (#D9063; Millipore Sigma: 20 times with pestle A and 20 times with pestle B)[41]. Undissociated tissue, largely composed of blood vessels, was removed by centrifugation at $200 \times g$ for 1.5 min at 4 °C[42], and the supernatant containing the nuclear material filtered through a 30 µm strainer and centrifuged at $500 \times g$ for 5 min at 4 °C. The resulting nuclear pellet was resuspended in nuclei lysis EZ buffer, incubated on ice for 5 min, washed by centrifugation, and resuspended in 300 µl nuclei EZ storage buffer by gentle trituration with a micropipette. From the total resuspended pellet volume, 10% was reserved as input nuclei fraction and the rest was diluted with 1.6 ml nuclei purification buffer (NPB: 20 mM HEPES, 40 mM NaCl, 90 mM KCl, 2 mM EDTA, 0.5 mM EGTA, 1X Halt[TM] protease inhibitor cocktail), and subjected to the INTACT protocol[10]. Briefly, 30 µl of resuspended M-280 Streptavidin Dynabeads (#11205, ThermoFisher Scientific) were added into a fresh 2 ml microcentrifuge tube and washed with 1 ml of NPB using a DynaMag-2 magnet (#12321; ThermoFisher Scientific) for a total of three washes (1 min incubation/each). The washed beads were reconstituted to their initial volume (30 µl) with NPB and gently mixed with the nuclear suspension. The mixture of nuclei and magnetic beads was incubated at 4 °C for 40 min under gentle rotation settings to allow the affinity binding of streptavidin beads to the cell-specific, biotinylated nuclei. After incubation, the streptavidin-bound nuclei were magnetically separated with the DynaMag-2 magnet for a period of 3 min and the unbound nuclei collected in a fresh 2 ml microcentrifuge tube, centrifuged at 4 °C ($1000 \times g$, 3 min), resuspended in 100 µl of NPB and reserved as the negative nuclei fraction. The nuclei bound to the beads were washed in the magnet for three washes (1 min/each), resuspended in 30 µl of NPB, and reserved as the positive nuclei fraction. From each nuclear fraction [input, negative (depleted of biotinylated nuclei), and positive (enriched in biotinylated nuclei)], a 3 µl aliquot was mixed with equal volume of DAPI counterstain and used for confocal microscopy visualization and calculation of purity percentage (3–5 fields of view per sample). The AllPrep DNA/RNA kit Micro (#80284, Qiagen, Germantown, MD) was used to extract gDNA from each sample[10]. gDNA was quantified with a Nanodrop 2000c spectrophotometer (Thermofisher Scientific) and its quality assessed by genomic DNA D1000 (#5067–5582) with a 2200 Tapestation analyzer (Agilent Technologies, Santa Clara, CA).

**Bisulfite amplicon sequencing (BSAS).** INTACT-isolated gDNA samples (input, negative fraction, and positive fraction) and mouse methylation controls (#80–8063-MGHM-5 and #80–8064-MGUM-5; EpigenDX, Hopkinton, MA) were diluted in nuclease free elution buffer (Qiagen) to a 10 ng/µl concentration (200 ng gDNA in 20 µl final volume). DNA was bisulfite converted for methylation analysis with the EZ DNA Methylation-Lightning[TM] Kit (#D5030T; Zymo Research, Irvine, CA), according to the manufacturer's guidelines. For methylation quantitation of gene promoters, primer sets (Integrated DNA Technologies; Supplementary Data 4) were designed based on the appropriate National Center for Biotechnology Information (NCBI) reference genome using the Methyl Primer Express v1.0 software (Thermofisher Scientific) to amplify 250–350 bp regions of interest

upstream or downstream the transcription start site (TSS) from bisulfite-converted DNA[43]. Bisulfite specific PCR optimization protocols were run to amplify and visualize amplicons by HSD1000 Tapestation. PCR amplicons were cleaned with Agencourt AmpureXP beads (#A63882; Beckman Coulter Life Sciences, Indianapolis, IN) using a two-sided size selection with 0.7× bead ratio followed by 0.15× bead ratio. Following clean-up, the amplicons were quantified using Qubit[TM] dsDNA HS assay kit (#Q32851; Thermofisher Scientific) and 5 ng of each amplicon was pooled per sample. One ng of the pooled amplicons was used for library construction with the Nextera XT DNA library preparation kit (#FC-131–1096; Illumina, San Diego, CA), according to the manufacturer's guidelines. Libraries were quantified with Qubit[TM] dsDNA HS assay kit and TapeStation HD1000, normalized to 1 nM or 4 nM, and pooled for sequencing. Pooled libraries were then sequenced on iSeq or MiSeq (Illumina) at loading concentrations 35 pM or 8 pM, respectively. Fastq files were aligned to amplicon sequences in CLC Genomics Workbench 11.0 (Qiagen) using the "Map Bisulfite Reads to Reference" feature. Site-specific CpG (CG) and CH methylation percentages were extracted for downstream analysis.

**Library construction and oxidative bisulfite sequencing (OxBS-seq).** For each input, negative, and positive INTACT-isolated sample 400 ng of gDNA was brought to 50 µl volume with 1× low-EDTA TE buffer and sheared with a Covaris E220 sonicator (Covaris, Inc., Woburn, MA) to an average 200 base pair size using the following settings: intensity of 5, duty cycle of 10%, 200 cycles per burst, two cycles of 60 seconds, at 7 °C. The size of sheared products was confirmed by capillary electrophoresis (DNA D1000, Agilent). gDNA fragments were cleaned by an Agencourt bead-based purification protocol, after which gDNA was quantified (Qubit[TM] dsDNA, Thermofisher Scientific). Two aliquots of 200 ng gDNA fragments were prepared in a 12 µl volume to which 1 µl of spike-in control DNA (0.08 ng/ul) with known levels of specific mC, hmC, and fC at individual sites was added. End repair, ligation of methylated adaptors (#L2V11DR-BC 1–96 adaptor plate, NuGEN, Tecan Genomics, Inc., Redwood City, CA) and final repair were performed according to manufacturer's instructions (Ovation Ultralow Methyl-Seq Library System, NuGEN)[14]. Of the two DNA aliquots per sample, one was oxidized and then bisulfite- converted and the other only bisulfite-converted with the True Methyl oxBS module (NuGEN) with desulfonation and purification. 22 µl of libraries were eluted from the magnetic beads. qPCR was used to determine the number (N) of PCR cycles required for library amplification. Bisulfite-converted samples were amplified for seven cycles while oxidative bisulfite- converted samples were amplified for 11 cycles [95 °C- 2 min, N (95 °C-15 s, 60 °C-1 min, 72 °C-30s)]. Amplified libraries were purified with Agencourt beads and eluted in low-EDTA TE buffer. TapeStation HD1000 was used to validate and quantify libraries. Amplified libraries were normalized to a concentration of 4 nM and pooled, denatured, and diluted to 12 pM for sequencing on the NovaSeq 6000 (Illumina) according to manufacturer's guidelines with the exception of a custom sequencing primer (MetSeq Primer) that was spiked in with the Illumina Read 1 primer to a final concentration of 0.5 µM.

**OxBS-seq data analysis.** Global levels of mCG, hmCG, and mCH were analyzed[14]. Prior to alignment, paired-end reads were adaptor-trimmed and filtered using Trimmomatic[44] 0.35. End-trimming removed leading and trailing bases with Q-score<25, cropped four bases from the start of the read, dropped reads less than 25 bases long, and dropped reads with average Q-score<25 Alignment of trimmed bisulfite-converted sequences was carried out using Bismark[45] 0.16.3 with Bowtie 2[46] against the mouse reference genome (GRCm38/mm10). Bams were deduplicated using Bismark. Methylation call percentages for each CpG and non-CpG (CH) site within the genome were calculated by dividing the methylated counts over the total counts for that site in the oxidative bisulfite-converted libraries (OXBS). Genome-wide CpG and CH methylation levels were calculated separately. Hydroxymethylation levels in CpG (hmCG) and CH (hmCH) contexts were calculated by subtracting call levels from the oxidative bisulfite-converted (OXBS) libraries from the bisulfite-converted (BS) libraries. BAM files generated by MethylSeq (Basespace, Illumina) were run through MethylKit in R[47] to generate context-specific (CpG/CH) coverage text files. Bisulfite conversion efficiency for C, mC, and hmC was estimated using CEGX spike-in control sequences. Untrimmed fastq files were run through CEGX QC v0.2, which output a fastqc_data.txt file containing the conversion mean for C, mC, and hmC. Analysis of methylation levels in the proximity of the promoter region was performed on a list of selected genes as follows. The R package Enriched Heatmap[48] was used to intersect methylation call files with genomic coordinates of gene lists. Flanking regions of 4000 nucleotides were constructed upstream of the transcription start site (TSS) and downstream of the transcription end site (TES) and then split into 20 bins of 200 nucleotides each. The gene body was split into 27 equal bins, depending on the gene length. The average of each bin for all genes in the list was then plotted versus the bin number to give a visualization of the overall pattern of mCG within and around the genes contained in the gene lists. Average mCG and hmCG levels were calculated for the upstream region (−4kb to TSS), gene body (TSS to TES), and downstream region (TES to +4 kb) for each gene list and biological replicate, and subjected to 2-way ANOVA statistical analysis with Sidak's multiple comparisons correction (GEO repository under accession code GSE140271).

Repeat element mCG, mCH, and hmCG was also examined. Repeat masker bed files were extracted from the UCSC Genome Browser Table Browser[49]. The context-specific CpG/CH MethylKit text files were intersected with the repeat masker bed files using 'bedtools', and percent methylation was calculated by dividing the average percent methylation at all common sites by the total number of sites. This was done for long interspersed nuclear elements (LINE), short interspersed nuclear elements (SINE), long terminal repeats (LTR), and simple repeats.

**Translating ribosome affinity purification (TRAP) and RNA extraction**. The purification of cell-specific RNA from brain tissue from Tam-induced Aldh1l1-cre/ERT2+; NuTRAP+ and Cx3cr1-cre/ERT2+; NuTRAP+ mice was achieved by following an established protocol, with slight modifications[11,50,51]. For each mouse, a hemisected half-brain or hippocampus was minced into small pieces and homogenized in 2 ml ice-cold homogenization buffer (50 mM Tris, pH 7.4; 12 mM MgCl2; 100 mM KCl; 1% NP-40; 1 mg/ml sodium heparin; 1 mM DTT) supplemented with 100 µg/mL cycloheximide (#C4859–1ML, Millipore Sigma), 200 units/ml RNaseOUT™ Recombinant Ribonuclease Inhibitor (#10777019; Thermofisher), and 1× cOmplete™, EDTA-free Protease Inhibitor Cocktail (#11836170001; Millipore Sigma) with a glass dounce tissue grinder set (#D8938; 10 times with pestle A and 10 times with pestle B). Homogenate was transferred to a 2 mL round-bottom tube and centrifuged at 12,000 × g for 10 min at 4 °C. After centrifugation, 100 µL of the supernatant was saved as input. The remaining supernatant was transferred to a 2 mL round-bottom tube and incubated with 5 µg/µl of anti-GFP antibody (ab290; Abcam) at 4 °C with end-over-end rotation for 1 h or overnight. Dynabeads™ Protein G for Immunoprecipitation (#10003D; Thermofisher) were washed three times in 1 ml ice-cold low-salt wash buffer (50 mM Tris, pH 7.5; 12 mM MgCl2; 100 mM KCl; 1% NP-40; 100 µg/ml cycloheximide; 1 mM DTT). After removal of the last wash, the homogenate/antibody mixture was transferred to the 2 ml round-bottom tube containing the washed Protein-G Dynabeads and incubated at 4 °C with end-over-end rotation for an additional 2 h. Magnetic beads were collected using a DynaMag-2 magnet and the unbound-ribosomes and associated RNA saved as the "negative" fraction (depleted). Beads were then washed three times with 1 ml of high-salt wash buffer (50 mM Tris, pH 7.5; 12 mM MgCl2; 300 mM KCl; 1% NP-40; 100 µg/ml cycloheximide; 2 mM DTT). Following the last wash, 350 µL of Buffer RLT (Qiagen) supplemented with 3.5 µl 2-β mercaptoethanol was added directly to the beads and incubated with mixing on a ThermoMixer (Eppendorf) for 10 min at room temperature. The beads were magnetically separated and the supernatant containing the target bead-bound ribosomes and associated RNA was transferred to a new tube. 350 µl of 100% ethanol was added to the tube ("positive" fraction: enriched in transcriptome associated to EGFP-tagged ribosomes) and then loaded onto a RNeasy MinElute column. RNA was isolated using RNeasy Mini Kit (#74104, Qiagen), according to manufacturer's instructions. RNA was quantified with a Nanodrop 2000c spectrophotometer (Thermofisher Scientific) and its quality assessed by HSRNA screentape with a 2200 Tapestation analyzer (Agilent Technologies).

**Library construction and RNA sequencing (RNAseq)**. The NEBNext Ultra II Directional Library Prep Kit for Illumina (#NEBE7760L; New England Biolabs Inc., Ipswich, MA) was used on 25 ng of total RNA for the preparation of strand-specific sequencing libraries from each TRAP-isolated RNA sample (input, negative fraction, and positive fraction) and from conventionally isolated RNA samples from brain (tissue), according to manufacturer's instructions. Briefly, polyA containing mRNA was purified using oligo-dT attached magnetic beads. mRNA was chemically fragmented and cDNA synthesized. For strand-specificity, the incorporation of dUTP instead of dTTP in the second strand cDNA synthesis does not allow amplification past this dUTP with the polymerase. Following cDNA synthesis, each product underwent end repair process, the addition of a single 'A' base, and finally ligation of adapters. The cDNA products were further purified and enriched using PCR to make the final library for sequencing. Library sizing was performed with HSRNA ScreenTape (#5067–5579; Agilent Technologies) and libraries were quantified by qPCR (Kappa Biosystems, Inc., Wilmington, MA). The libraries for each sample were pooled at 4 nM concentration and sequenced using an Illumina NovaSeq 6000 system (SP PE50bp) at the Oklahoma Medical Research Foundation Genomics Facility.

**RNAseq data analysis**. Following sequencing, reads were trimmed, aligned, differential expression statistics and correlation analyses were performed in Strand NGS software package (Agilent)[14]. Reads were aligned against the Mm10 build of the mouse genome (2014.11.26). Alignment and filtering criteria included: adapter trimming, fixed 2 bp trim from 5' and 6 bp from 3' ends, a maximum number of one novel splice allowed per read, a minimum of 90% identity with the reference sequence, a maximum of 5% gap, trimming of 3' end with Q < 30. Alignment was performed directionally with Read 1 aligned in reverse and Read 2 in forward orientation. Reads were filtered based on the mapping status and only those reads that aligned normally (in the appropriate direction) were retained. Normalization was performed with the DESeq algorithm[52]. Transcripts with an average read count value >20 in at least 100% of the samples in at least one group were considered expressed at a level sufficient for quantitation per tissue and those

transcripts below this level were considered not detected/not expressed and excluded, as these low levels of reads are close to background and are highly variable. For statistical analysis of differential expression, a one-way ANOVA or two-way ANOVA with the factors of TRAP fraction and treatment and a Benjamini–Hochberg Multiple Testing Correction followed by Student–Newman Keuls post-hoc test were used. For those transcripts meeting this statistical criterion, a fold change > |1.25| cutoff was used to eliminate those genes, which were statistically significant but unlikely to be biologically significant and orthogonally confirmable due to their very small magnitude of change. Visualizations of hierarchical clustering and principle components analysis were performed in Strand Next Generation Analysis Software (NGS) (Version 3.1, Bangalore, India). The entirety of the sequencing data is available for download in FASTQ format from NCBI Sequence Read Archive (GSE140895 and GSE140974). Cell-type-specific maker gene lists were generated from the reanalysis published by McKenzie et al[18]. of immunopurified[53] and high throughput single-cell data from mice[54,55]. Published lists were filtered first by mean enrichment score of ≥3.5 and secondly to remove any genes that appeared on lists for multiple cell types. Comparisons of astrocyte gene enrichment in this study to previously published Aldh1l1-RiboTag[12] and Gfap-TRAP[19] were performed by downloading raw fastq files with GEO accession numbers GSE84540 and GSE99791, respectively, and processing the files through StrandNGS as above, with minor alterations as necessitated by the type of sequencing data. After alignment, astrocyte markers were classified by differential expression between the input and positive fractions of Aldh1l1-RiboTag, Aldh1l1-TRAP, and Aldh1l1-NuTRAP was assessed by t-test, BHMTC < 0.05 and FC > 5. The intersection of these gene lists was then used to construct a ribosomal-tagging astrocyte gene list. In a similar manner, microglial marker genes identified in this study (t-test, BHMTC p < 0.05 and FC(pos/input)>5) were compared to Cx3cr1 (Jung)-RiboTag[5] and Cx3cr1(Litt)-TRAP[3] by downloading raw fastq files with GEO accession numbers GSE114001 and GSE108356, respectively, and processing as above. Gene expressions of selected genes from previously published gene lists[12,19,20] were imported into the IPA software Ingenuity Pathway Analysis (IPA) 01.12 (Qiagen Bioinformatics) to assess pathway/biological function enrichment analysis.

**Quantitative PCR (qPCR)**. Confirmation of gene expression levels was performed with qPCR[14,56,57]. cDNA was synthesized with the ABI High-Capacity cDNA Reverse Transcription Kit (Applied Biosystems Inc., Foster City, CA) from 25–100 ng of purified RNA. qPCR was performed with gene-specific primer probe fluorogenic exonuclease assays (TaqMan, Life Technologies, Waltham, MA, Supplementary Table 1) and the QuantStudio™ 12 K Flex Real-Time PCR System (Applied Biosystems). Relative gene expression (RQ) was calculated with Expression Suite v 1.0.3 software using the $2^{-\Delta\Delta Ct}$ analysis method with Hprt or Gapdh as an endogenous control.

**Stable isotope labeling**. Microglial proliferation was measured as incorporation of deuterium into purine deoxyribose[58]. Briefly, mice were given an intraperitoneal injection of 99.9% D2O and subsequently provided drinking water enriched with 8% D2O for 30 days. Following INTACT isolation, DNA was extracted from nuclei using QiAamp DNA mini kit (Qiagen, Valencia, CA) according to manufacturer protocol. Extracted DNA was hydrolyzed overnight at 37 °C with nuclease S1 and potato acid phosphatase. Hydrolysates were prepared for analysis of the penta-fluorobenzyl-N,N-di(pentafluorobenzyl) derivative of deoxyribose by GC-MS. Enrichment of deuterium in DNA from bone marrow was similarly analyzed for each animal to determine precursor enrichment. Fraction of new DNA was calculated based on the product/precursor relationship.

**Statistics and reproducibility**. Datasets with groups of n < 10 were analyzed using GraphPad Prism software 8.2.0 (435) (San Diego, CA) and represented as dot plots with underlying bar graph with mean ± S.E.M. (standard error of the mean) or box plots consisting of median (boxes spanning Q1–Q3 and whiskers to the maximum and minimum value). Further information on research design is available in the Nature Research Reporting Summary linked to this article and in Supplementary Data 5.

**Reporting summary**. Further information on research design is available in the Nature Research Reporting Summary linked to this article.

## Data availability
Sequencing data that support the findings of this study have been deposited in GEO repository with the GSE140271 accession code for information on oxBS-seq data (used for Fig. 9 and Supplemental Figs. 13 and 14). The entirety of the RNA-sequencing data is available for download in FASTQ format from NCBI Sequence Read Archive (GSE140895, GSE159106, GSE140974). Supplementary imaging data are available from figshare.com with https://doi.org/10.6084/m9.figshare.12670895 and https://doi.org/10.6084/m9.figshare.12669698. All source data underlying the graphs presented in the main/Supplementary Figs. are available in Supplementary Data 7. Other data that support the findings of the study are available from the corresponding author (W.M.F.) upon request.

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

## Acknowledgements

We acknowledge the Laboratory for Molecular Biology and Cytometry Research at OUHSC for the use of the Core Facility, which provided flow cytometry services, the Cellular Imaging Core, at the Dean McGee Institute, OUHSC, for providing access to vibratome and confocal imaging equipment, and the Oklahoma Medical Research Foundation Clinical Genomics Center, which provided sequencing services. Computing

for this project was performed at the OU Supercomputing Center for Education & Research (OSCER) at the University of Oklahoma (OU). OMRF Animal Research Facility assisted with animal husbandry and OMRF Imaging Core Facility with imaging analyses. We also acknowledge Adeline Machalinski (mouse colony management and genotyping), Robyn Berent (administrative support and lab management), P. Marlow for figure preparation, and Dr. Michael H. Elliott for critical scientific discussions during the preparation of this manuscript. This work was supported by grants from the National Institutes of Health (NIH) P30AG050911, R56AG059430, R01AG052606, R01AG059430, R01AG069742, R00AG051661, P20GM125528, T32AG052363, F31AG064861, F31AG063493, P30EY021725, Oklahoma Center for Adult Stem Cell Research (OCASCR), a program of the Oklahoma Tobacco Settlement Endowment Trust, American Federation for Aging Research, BrightFocus Foundation, and Presbyterian Health Foundation. This work was also supported in part by the MERIT award I01BX003906 from the United States (U.S.) Department of Veterans Affairs, Biomedical Laboratory Research and Development Service.

## Author contributions

A.J.C.-E.: design of the work, execution of experiments, data acquisition, analysis, and interpretation, figure generation, manuscript writing and preparation. S.R.O.: design of the work, execution of experiments, data acquisition, analysis, and interpretation, figure generation, manuscript writing and preparation. D.R.S.: data analysis. V.A.A.: execution of experiments, data acquisition, and analysis. K.B.B.: execution of experiments, data acquisition, and analysis. H.P.: data analysis. N.L.E.: execution of experiments, data acquisition, and analysis. J.J.R.: execution of experiments, data acquisition, and analysis. A.L.S.: conceptual design of the study, data interpretation, manuscript writing. M.B.S.: conceptual design of the study, data interpretation, manuscript writing. M.J.B.: conceptual design of the study, data interpretation, manuscript writing. B.F.M.: conceptual design of the study, data interpretation, manuscript writing. A.R.: conceptual design of the study, data interpretation, manuscript writing. W.M.F.: Corresponding author, conceptual design of the study, data analysis and interpretation, figure generation, manuscript writing, preparation, and submission.

## Competing interests

The authors declare no competing interests.
