## [Peer Review File · Communications Biology]

Reviewers' Comments:

Reviewer #1:

Remarks to the Author:

In 'Inducible cell-specific mouse models for paired epigenetic and transcriptomic studies of 1 microglia and astroglia', Chucair-Elliott et al. describe mouse models that can be used to study gene expression and chromatin signatures from specific CNS cell types. They have generated these models to detect these features in both an astrocyte and microglia specific way through the combination of existing models (NuTRAP combined with an astrocyte specific Cre driver and a microglia specific Cre driver). So the technology is not novel, but here different elements are combined to allow for the implementation in CNS cell-types.

In the paper they demonstrate and validate the specificity of the generated mouse models and then go on to implement 1 of the models to demonstrate the applicability of the technology to detect expression and chromatin changes upon perturbation of the system. For this they model inflammation by ip administration of LPS and analyze the response in microglia.

Generally the implementation of the NuTRAP technology in the CNS is useful and the manuscript describes experiments that seem to be well controlled and executed. The manuscript does not describe any major biological conclusions.

Please find below some comments that would further strengthen and clarify the manuscript

1. The number of detected astrocyte enriched and microglia enriched genes that are detected with the NuTRAP seems to be on the low side (compared to similar Ribotag approaches and compared to data on hepatocytes in the original NuTRAP paper by Loh et al.). Can the authors comment on this? Is this related to filtering/thresholding? Was the sequencing depth sufficient?

2. The IF data presented in the supplemental information where the eGFP, mCherry and cell-specific markers are depicted is not so clear for both mouse lines that were used. Additional zoomed-in images would be useful. In addition, to better evaluate the degree to which the reporters are expressed in the same cells, the overlap between eGFP and mCherry should be presented in the FACS plots.

3. From the FACS plots in Figures 1 and 4 it is concluded that those data correspond to the percentages of astrocytes and microglia in the CNS, but that conclusion cannot be made. Only viable cells are analyzed, what if different cell types have different susceptibility/viability in response to the extraction procedure?

4. In line 114 it is written that complement is over-represented in astrocyte-enriched transcripts. In my opinion the heatmap presented in figure S3 shows the opposite

5. The marker genes depicted in figures 3 and 6 have some issues. Why are the genes selected and depicted not the same as for example the marker genes depicted in figure 2? For robustness, more than 1 gene should be analyzed and shown. Preferably with some more commonly used marker genes.

Fabp7 is not an exclusively astrocytic marker in the brain, it has been reported that approximately 50% of the Fabp7 positive cells in the CNS are not astrocytes, it is also expressed by OPCs (PMID 21938553). The expression of Fabp7 by non-astrocytes is confirmed by the expression data presented on brainrnaseq.org

Similarly Eno2 is not a marker exclusively expressed in neurons

Therefore more and better marker genes have to be included in Figure 3 (and/or supplemental)

6. Line 133: A minor trend... This has to be removed, there is no significant increase in methylation for Grp84 and Eno2

7. In line 174: Genes enriched in the... It is not completely clear which genes are meant here. The

- 295 identified by NuTRAP, the 142 ribosomal tagging common, or the 101 isolation method-independent?
8. How robust are the differences in hmCG between astrocytes and microglia? What was used for inter-sample normalization? Were for example spike-ins used for that purpose?
 9. The data presented in figure 7. Is there a significant correlation between the level of methylation and gene expression?
 10. The formulation in line 275-276 is a bit weird. Induction of ... genes is seen in LPS, but not input. Considering that induction is measured relative to input, it is also technically not possible to see induction in the input.
 11. In figure 8 there is a discrepancy between DNA methylation and gene expression. Can that be explained by the timing of sample collection? Or is there another good explanation
 12. For figure 8, some more genes should be analyzed. Including genes that do not show a response to LPS. And the expression values and methylation levels should be correlated. Genome-wide is perhaps not necessary, but 3 genes is very limited to draw robust conclusions.
 13. In the discussion cell-specific contributions to the transcriptome and epigenome are mentioned. However, single cell RNA-sequencing data from recent years have shown that astrocytes and microglia are not homogeneous cell populations. Do the authors see a contribution for their technology (possibly modified) for that?

Minor comments

1. Genes and gene products (including RNA molecules) need to be written in italic (now it is sometimes confusing since also proteins are described in the text)
2. There are quite a few small mistakes /typo's (some examples: line 58, microglial: the l should be deleted, line 109: studes should read studies, line 123: putatively should read putative, etc.)
3. Figure 2C, D and 5C, D the labeling (spelling) of categories on the x-axis of the violin and next to the heatmaps are inconsistent
4. In the methods, the vav-Cre is mentioned. And that it did not work, because that particular cre driver is not very specific, also not in these essays. This information is hidden away now, but if the authors think this is an important point to make, maybe this should also be briefly mentioned in the results section. Otherwise I would recommend to delete it

Reviewer #2:

Remarks to the Author:

Chucair-Elliott et al. set out to validate the use of the NuTRAP model to obtain astrocyte- and microglia-specific transcriptomic and epigenomic profiles. Using parallel Translating Ribosomal Affinity Purification for RNA-seq the authors confirm that the NuTRAP-enriched astrocyte and microglia populations show enrichment for their respective cell-type specific markers. The authors utilized INTACT-inspired streptavidin bead-mediated nuclear isolation to study the DNA methylation (i.e., 5-methylcytosine (5mC) and 5-hydroxymethylcytosine (5hmC) profiles of these cells, again showing de-enrichment, or enrichment, of DNA methylation at genes expected to be enriched, or de-enriched, in these cells. Finally, the authors focus on microglia while studying changes to their transcriptional and DNA methylation landscapes in response to LPS. In doing so, the authors reveal that NuTRAP-generated microglia DNA methylation profiles reveal dynamic changes in DNA methylation that cannot be detected with approaches that are cell-type agnostic.

The authors make a compelling case in support of using the NuTRAP model to investigate the transcriptional and epigenetic landscape of microglia and astrocytes. Yet, whereas the major thrust of the manuscript deals with the validation of NuTRAP, the extent of the insights gleaned from this technology, as relates to microglia and astrocyte biology, seem to be rather modest. Moreover, although evidence of the NuTRAP model being used to study CNS cellular population has not been

published to date, prior evidence of its use to study adipocytes and hepatocytes (PMC5291126, PMC5932137) has already removed much of the doubt that this technology could be applied to any cell-type. Nonetheless, the authors do conclusively demonstrate that this technique is a viable option for those wishing to study these two cellular populations. Furthermore, whereas other groups have used single-pronged approaches (TRAP- or INTACT-exclusive) to obtain transcriptomic (PMC5986066, PMC6242869, PMC6090564) and epigenomic (e.g., H3K27me3, PMC6090564; DNA methylation, PMC6242869) profiles from microglia, it does appear that this manuscript is the first to unveil in vivo astrocyte-specific DNA methylation profiles.

Major Issues:

1. The TRAP negative fraction is not truly a negative fraction, but rather a positive-depleted fraction. It's impossible to remove all positive nuclei with TRAP, or INTACT for that matter, and thus impossible to generate a true "negative" fraction. One would need to FACS and gate for the mCherry-negative population to obtain a true negative fraction. Call the negative fraction a depleted fraction, or something to that effect.
2. In figure 3 the authors show that ~90% of mcherry (astrocyte)-stained nuclei are bound to streptavidin. This result speaks to high affinity of streptavidin for biotinylated astrocyte nuclei, but the purification of the nuclei population in the positive fraction is unclear. Isolated nuclei are notoriously sticky, which raises the possibility that the NuTRAP/INTACT-selected astrocyte nuclei might be coupled to non-astrocyte nuclei. This would contribute to some degree of contamination. The authors should measure the percentage of mCherry-negative/strept. bead-bound nuclei in the "positive" fraction in figure 3A and 3B.
3. As mentioned above, the impact of the biological insights generated from this manuscript are not immediately clear.

Minor Issues:

1. Show the representative tracks for the WGoBS data.
2. Show the head maps for the LPS RNA-seq data.
3. Show Principle Components Analysis (PCA) for the WGoBS experiments.

Reviewer #3:

Remarks to the Author:

The authors combine a Cre- dependent mouse line, NuTRAP, previously developed by another group for adipocyte studies, with established Cre lines that are functional in brain cells. They find that the Cre lines were able to activate NuTRAP after giving the inducing chemical, tamoxifen. Once activated in cell types of interest, NuTRAP labels nuclei and ribosomes that can be purified for gene expression studies. The authors study two interesting cell types, astro- and microglia. The authors present data from both domains, with the transcriptome data being somewhat expected, and the DNA methylation data being more novel.

Despite the overall positive impression of the manuscript I have some suggestions and questions for the authors.

- 1) The sentence beginning on line 200 claims:

"Moreover, no studies have compared the methylome of different cell types, such as astrocytes and microglia, using a combination of inducible cre-recombinase and NuTRAP technologies."

Also, in the sentence beginning on 331 claims:

"In summary, the results offer extensive evidence to support the combination of inducible cre/lox and NuTRAP models as a suitable and powerful approach for the parallel study of the cell-specific epigenetic and transcriptomic signatures in the brain."

Also, line 383:

"Collectively, our experiments demonstrated that the NuTRAP approach can be applied to CNS cell populations and..."

There are two issues with these and similar claims.

First, these claims could leave readers with the impression that this manuscript represents the first report combining epigenetic and transcriptomic signatures of specific cell types of the mouse brain. In fact, work with the original TRAP (Mellen et al., 2012) as well as INTACT (Mo et al., 2015, yes DNA and RNA were studied) and Tagger (Kaczmarczyk et al., 2019) all did this previously. The manuscript would be greatly improved if the claims of technical novelty were replaced with a comparison of all relevant published methods and with a focus on the new biological data, specifically the methylation studies.

Second, please explain the value of the temporal control for NuTRAP and why this is an important development that warrants being repeated throughout the manuscript. What is gained by turning the system on after development? Is the construct toxic if expressed during development? Probably not. The authors write that it might be detrimental to express cre through development but they offer no evidence that this is a problem that requires limited cre activity, not for CNS cells including the glia types studied in this report. The only apparent need for the demonstrated experiments is that it reduces the activation of off-target cells. This is why Cre-ERT2 lines are typically used for glia. In other words, the value in using a tamoxifen Cre line was to control the off targeting of cells by the Cre (the shortcomings of Cre) not due to a problem with constitutive expression of NuTRAP. A better explanation of the value of the temporal control as it relates to the current data is needed. Otherwise, this is simply a technical demonstration that the system can be induced and the text should be accordingly changed. In this case, the main result was that the system was employed to study brain glia. A similar mouse line expressing multiple proteins to study gene expression called Tagger was used to study brain neurons, so the ability to study glia is not unexpected.

2) Another problem is the phrase "paired analysis" implies that both types of data are generated from the same sample. However, it is apparent that INTACT and TRAP were not done on the same samples since the homogenates were made with different buffers. Perhaps one hemisphere was used for one method and the second hemisphere was used for the other method? The phrase "in parallel" was used in places but this does not mean they were from the same brain. That should be clarified for the interpretation of the presented data. Also, since different buffers were used for the different methods, the workflows in supplemental figures 1 and 10 are very misleading and must be corrected. They imply that the same homogenate was split for the different methods but this is not true.

3) More importantly, the authors are obtaining only one level of gene expression from each lysate. The method would be much more useful if getting multiple domains from the same homogenate. This was accomplished with the other methods. Nuclei and TRAP should be done on the same homogenate, or at the very least a protocol should be established where that could be done. If this was attempted but without success then that information should be clearly noted in a prominent location (e.g. discussion) of the manuscript. This is the most valuable purpose for these types of models (NuTRAP and Tagger) and readers will undoubtedly want to know this.

4) NGS samples were purified from hemibrains but the histology only shows hippocampal areas. The inclusion of images of the whole brain should therefore be displayed. Any images that include only a specific region (e.g. figures 1c and 4c) should have that region clearly labeled ("Brain" is not a sufficient label.) Readers will want to know if the activation driven by these cre lines worked as well for NuTRAP as they did for RiboTag. Any differences would be useful information to researchers designing experiments, as well as readers of this area of the scientific literature.

5) The FACS studies applied gating windows that excluded certain populations of cells that appeared to be GFP+ (dots above the gate windows in the main text figures). It seems reasonable that these cells were expressing GFP and TRAP, and would therefore be in the pool of captured nuclei and RNA. Indeed, in supplemental figure 4 B a'' there is prominent GFP staining of a cell sized and shaped more like a neuron than microglia. What was the rationale for not including an analysis with all the GFP+ cells in the FACS analysis? That is, how were the dots above the gating windows excluded? Furthermore, why not also do an experiment where the first gate is for the cell marker (e.g. cd11b) then determine how many of those are expressing GFP? It seems the purpose of the FACS study should have been to determine the completeness and specificity of the proportion of GFP+ cells in the different glia, but the data are not presented in a way that addresses these important questions. Also, a definition of PE on the Y-axis should be included.

The NGS analysis was well done with some new and interesting information. The writing was clear and the explanations of some complicated data analyses were also well done. The new biological insights on methylation are more noteworthy than the technical advances.

We thank the Editor and Reviewers for their time and interest in considering our original work for publication in Communications Biology. Finding ourselves in the middle of the Covid-19 pandemic, and with the limitations that this emergency situation causes to the scientific community, we have made every effort to address the Reviewers' comments. We include significant amounts of new imaging, RNA-Seq, qPCR, and BSAS data and have revised the text, analyses, and figures. These data serve to strengthen the conclusion that the Aldh111- and Cx3cr1- Cre-ERT2/NuTRAP models are robust, validated tools to be applied in cell-specific epigenetic and transcriptomic studies of the brain and that this approach can be applied in general to neuroscience studies. Alterations to the text, as outlined below are denoted in **red text** in the manuscript.

Reviewers' comments:

Reviewer #1 (Remarks to the Author):

1. The number of detected astrocyte enriched and microglia enriched genes that are detected with the NuTRAP seems to be on the low side (compared to similar Ribotag approaches and compared to data on hepatocytes in the original NuTRAP paper by Loh et al.). Can the authors comment on this? Is this related to filtering/thresholding? Was the sequencing depth sufficient?

Response: We have investigated this question through a number of approaches.

1) The filtering/thresholding of the data sets was as similar as possible for the between-studies comparison. Raw data from these studies were re-aligned to the same genome build and subjected to the same statistical and fold changes cutoffs with the exceptions noted below. Differences in gene expression enrichment could be potentially explained by the use of different cre-expressing mouse lines, isolation construct (NuTRAP, TRAP, and RiboTag), brain region(s) used in each study, number of samples used (both positive fractions and inputs), the technicalities associated with ribosomal isolation protocols, sequencing platform used, read length, depth of coverage, number of aligned reads, and statistical tests (e.g., the Boisvert study where a z-test was used due to n=1/positive fraction). The details of our own methodology and data analysis, and details available from the published studies used for comparison with are summarized in following two tables. As can be seen there are enough differences in all these variables that these are not strictly parallel studies, and we have noted that in the text that the direct comparisons of the numbers of enriched genes should be done cautiously.

Figure 2	Aldh111-RiboTag Srinivasan et al	Gfap-RiboTag Boisvert et al	Aldh111-NuTRAP Present Study
Cre mouse line	Aldh111 ^{cre} /ERT2 (#029655, Jax)	Gfap ^{cre} (#012886, Jax)	Aldh111 ^{cre} /ERT2 (#029665, Jax)
Reporter mouse line/affinity protocol	RiboTag (#011029, Jax)/TRAP [anti-HA antibody (#MMS-101R, Covance) + dynabeads Protein G (100.04D, ThermoFisher)]	RiboTag (#011029, Jax)/TRAP [anti-HA antibody (#3724, SCT) + Protein G magnetic beads (#88847, ThermoFisher)]	NuTRAP (#029899, Jax)/TRAP [anti-GFP antibody (#ab290, abcam) + dynabeads Protein G, (#10003D; ThermoFisher)]
Brain Region	Cortex	Cortex	Hemisected brain
Number and type of Input samples	4 Cleared homogenate	3 Cleared homogenate	3 Cleared homogenate
Number of "Positive" samples	4	1	3
Single end or paired end	Illumina NextSeq 500 PE75bp	Illumina HiSeq 2500 SR50bp	Illumina NovaSeq 6000 PE50bp
GEO Accession	GSE84540	GSE99791	GSE140895
Number of aligned reads	89,182,744 +/- 15,304,904	35,991,054 +/- 2,949,846	29,946,637 +/- 4,969,809
% of enriched genes observed in ≥1 other study	47%	89%	85%

Figure 5	Cx3cr1 ^{Litt} -RiboTag Ayata et al	Cx3cr1 ^{Jung} -TRAP Haimon et al	Cx3cr1 ^{Jung} -NuTRAP Present Study
Cre mouse Line	Cx3cr1 ^{cre/ERT2} (#021160, Jax)	Cx3cr1 ^{cre/ERT2} (#020940, Jax)	Cx3cr1 ^{cre/ERT2} (#020940, Jax)
Reporter mouse line/affinity protocol	EEF1A1-LSL.EGFP.L10 (J. Friedman (Rockefeller University, NY) /TRAP [anti-GFP antibody (19C8 and 19F7, Antibody & Bioresource Core Facility, Memorial Sloan-Kettering, NY) + biotinylated Protein L (GenScript, Piscataway, NJ) + Streptavidin MyOne T1 Dynabeads(Invitrogen)]	RiboTag (#011029, Jax) /TRAP [anti-HA antibody (#H9658, Sigma) + dynabeads Protein G (ThermoFisher)	NuTRAP (#029899, Jax)/TRAP [anti-GFP antibody (#ab290, abcam) + dynabeads Protein G, (#10003D; ThermoFisher)]
Brain Region	Striatum	Hemisected brain	Hemisected brain
Number and type of Input samples	2 Cleared homogenate	3 Cleared homogenate	6 Cleared homogenate
Number of "Positive" samples	2	3	6
Single end or paired end	HiSeq 2000 SR50bp	Illumina NextSeq 500 SR75bp	Illumina NovaSeq 6000 PE50bp
GEO Accession	GSE108356	GSE114001	GSE140895
Number of aligned reads	40,781,317+/-8,179,047	1,450,441 +/-367,368	30,451,921 +/- 3,879,410
% of enriched genes observed in ≥1 other study	55%	62%	85%

2) We chose as criteria for a cell type marker to have statistical significance and a fold change >5 enrichment to identify genes highly restricted in expression. While some differences are evident we have added Supplemental Figures 4 and 10 that show that while the set of genes that meet this arbitrary criteria differ somewhat they demonstrate a high level of agreement with cell sorting marker list and overall enrichment. These differences are likely due to the many experimental differences noted above.

3) We have also added pairwise correlations to Figures 2f and 5f. We have also altered Figure 2g and 5g to include 'markers' from 2 or more studies and a greater degree of overlap with cell sorting studies is evident. It should also be noted that the NuTRAP lines demonstrated a higher level of concurrence (85%) in marker genes found in at least one other study. The study for each cell type (Srinivasan or Ayata) with the most enriched genes demonstrated the lowest level of concurrence with other studies.

We include references to this new data in the text and also provide a discussion point that there is a great deal of concurrence between these models and approaches. The NuTRAP having the added benefit of not requiring an additional mouse line to perform INTACT and epigenetic analyses as compared to the TRAP or RiboTag lines.

2. The IF data presented in the supplemental information where the eGFP, mCherry and cell-specific markers are depicted is not so clear for both mouse lines that were used. Additional zoomed-in images would be useful. In addition, to better evaluate the degree to which the reporters are expressed in the same cells, the overlap between eGFP and mCherry should be presented in the FACS plots.

Response: In the revised version of this manuscript, 2X-zoomed insets of the originally provided images are included in Supplemental Figures 3, 7, 17, in order to clarify the eGFP, mCherry, and cell-specific marker expression co-localization. In addition, we have included new confocal images of representative areas of cortex, cerebellum, and hippocampus (Supplementary Figures 2 and 6), and made available for the Reviewers high resolution tile scan images through two private links (<https://figshare.com/s/fc817f9235f4384c5fb8> for Aldh111-NuTRAP and <https://figshare.com/s/de4f6fcb1c8fc858de10> for Cx3cr1-NuTRAP) covering the entirety of a sagittal brain section for both mouse lines.

We agree with the Reviewer in that assessing the overlap of eGFP and mCherry expression by flow cytometry (FC) analysis would be an additional measure to complement our observations by immunohistochemistry that clearly show overlapping of mCherry and GFP on recombined astrocytes and microglia (Figures 1 and 2,

Supplementary Figures 2, 3, 6, and 7). We would like to emphasize for the flow experiments in this study, the gating strategy used EGFP signal (and not mCherry signal) to assess that cre-recombination occurred in the expected cell type. We are confident that the flow data are clear and sufficient to support cell-specificity of cre-recombination in astrocytes in the Aldh1l1-cre/ERT2⁺; NuTRAP⁺ model and in microglia in the Cx3cr1-cre/ERT2⁺; NuTRAP⁺, upon tamoxifen (Tam) treatment. The principle of cre-mediated recombination in the presence of the NuTRAP construct, with expression of both mCherry and EGFP by the same cell was first demonstrated by Roh. et al¹, and is successfully reproduced in our hands. We collectively show consistent co-localization of EGFP and mCherry to the same cell in both Aldh1l1-cre/ERT2⁺; NuTRAP⁺ and Cx3cr1-cre/ERT2⁺; NuTRAP⁺ models by IHC (Figures 1, 2, Supplementary Figures 2, 3, 6, 7 and 16), and the agreement between TRAP-RNAseq profiles from EGFP-labeled polysomes and the methylation profiles of genomic DNA from biotinylated nuclei isolated by INTACT (Figures 2, 3, 5, 6, 7 and 8). It is also worth noting that the source of DNA and RNA materials for sequencing library preparation in our work is freshly isolated tissue and not sorted cells, and the FC experiments shown here do not define the outcome of TRAP and INTACT isolations.

3. From the FACS plots in Figures 1 and 4 it is concluded that those data correspond to the percentages of astrocytes and microglia in the CNS, but that conclusion cannot be made. Only viable cells are analyzed, what if different cell types have different susceptibility/viability in response to the extraction procedure?

Response: We realize that our wording here was a bit confusing in respect to the percentage of cells characterized as astrocytes and microglia by our flow cytometry approaches. The intent was not to state a conclusion of our study, but that these data agree with widely accepted cell population proportions in the brain. According to the literature, the cell constituency of astrocytes and microglia is accepted to be around 10-20%^{2, 3} and 5-10%⁴ respectively in the rodent brain. The percentages of the cell populations gated as EGFP⁺ ACSA2⁺ in Figure 1 and EGFP⁺CD11b⁺ in Figure 4 were within the expected “consensus cellularity” of the brain (regardless the technique and extraction procedures). These values are used only as a general positive control in that if wildly different proportions were observed that would be a cause for concern.

4. In line 114 it is written that complement is over-represented in astrocyte-enriched transcripts. In my opinion the heatmap presented in figure S3 shows the opposite.

Response: We have re-written the sentence to clarify the observation. The sentence has been reworded to express: “Astrocyte enriched transcripts further demonstrated over-representation of genes critical in astrocyte physiological functions⁵⁻⁷ such as cholesterol synthesis and transport, fatty acid metabolism, receptors/channels, and synapse modification (formation, function, and elimination), while under-representation of complement/immune mediators, commonly associated with microglial function (Supplementary Figure 5).”

5. The marker genes depicted in figures 3 and 6 have some issues. Why are the genes selected and depicted not the same as for example the marker genes depicted in figure 2? For robustness, more than 1 gene should be analyzed and shown. Preferably with some more commonly used marker genes. Fabp7 is not an exclusively astrocytic marker in the brain, it has been reported that approximately 50% of the Fabp7 positive cells in the CNS are not astrocytes, it is also expressed by OPCs (PMID 21938553). The expression of Fabp7 by non-astrocytes is confirmed by the expression data presented on brainrnaseq.org. Similarly, Eno2 is not a marker exclusively expressed in neurons. Therefore more and better marker genes have to be included in Figure 3 (and/or supplemental).

Response: We have significantly expanded the repertoire of gene markers analyzed for promoter methylation by BSAS that substantiate the conclusions of Figures 3 and 6 and are consistent with the panel of marker genes depicted in Figures 2 and 4 (all genes targeted for BSAS have been assayed for gene expression levels by qPCR). The selection of astrocytic markers assayed for BSAS include: Aldh1l1, Gfap, Fabp7, and Kcnj10. Microglial genes analyzed for BSAS were Cx3cr1, C1qa, Gpr84, and Aif1. We include BSAS data for the neuronal markers Eno2, Syt2, and Syt4.

In regard to the selection of Fabp7 as target for BSAS, our TRAP-RNA-seq data sets showed high enrichment in the positive fraction of Aldh1l1-cre/ERT2⁺; NuTRAP⁺ brains, in agreement with the high gene expression enrichment of Fabp7 represented in the Barres lab website <https://www.brainrnaseq.org/> (~1500 FPKM in

astrocytes vs 10-300 FPKM in other cell types (based on sorted-cell studies that are assumed to carry cell contaminations in the isolations). Other groups using sorted-cells⁸ and TRAP protocols^{5, 7} have reported enrichment of Fabp7 gene expression in astrocytes of the brain. On the other hand, the 2011 Histochem Cell Biol (PMID: 21938553) study noted by the Reviewer, falls short in demonstrating undebatable cell-specific localization of Fabp7 in the brain, as it relies on antibody staining and imaging techniques using a very limited number of markers (NG2, PDGFR α , GFAP, FABP7) and does not provide absolute quantitation in the whole tissue to support the conclusions.

6. Line 133: A minor trend... This has to be removed, there is no significant increase in methylation for Grp84 and Eno2

Response: This sentence has been removed from the revised version of this manuscript.

7. In line 174: Genes enriched in the... It is not completely clear which genes are meant here. The 295 identified by NuTRAP, the 142 ribosomal tagging common, or the 101 isolation method-independent?

Response: The statement refers to the gene enrichment of our study (NuTRAP) without comparing with other groups. We have edited the sentence to read: "Genes enriched in the microglia transcriptome in our study included an overrepresentation of genes regulated by PU.1 (also known as Spi1), a transcription factor that shapes the homeostatic functions of microglia (Supplementary Figure 8)."

8. How robust are the differences in hmCG between astrocytes and microglia? What was used for inter-sample normalization? Were for example spike-ins used for that purpose?

Response: Spike-in control sequences with known levels of base-specific methylation and hydroxymethylation were included in all bisulfite- (BS) and oxidative bisulfite- (OxBS) whole genome libraries. The conversion efficiencies are provided in Supplementary Figure 12. All libraries were prepared and sequenced at the same time to avoid batch effects. We see no difference in mC or hmC conversion between the Aldh1l1-NuTRAP and Cx3cr1-NuTRAP input DNA.

9. The data presented in figure 7. Is there a significant correlation between the level of methylation and gene expression?

Response: The present resubmission of this manuscript includes new paired BSAS and qPCR to assess the correlation between CG methylation (% mCG) and gene expression (normalized RQ) for cell specific targets, as part of Figure 8.

10. The formulation in line 275-276 is a bit weird. Induction of ... **genes is seen in LPS, but not input.** Considering that induction is measured relative to input, it is also technically not possible to see induction in the input.

Response: The idea behind that statement is that the magnitude of the effects of LPS up-regulating the expression of inflammation genes in the brain by qPCR is more pronounced when comparing positive fractions of LPS- vs PBS-treated mice than when comparing input fractions of LPS- vs PBS- treated mice. The interpretation is that that by enrichment for the transcriptome of a small cell population of the brain, such as microglia, the differences in inflammatory responses by microglia that are otherwise "diluted" in the input transcriptome become more pronounced. We have edited the sentence to clarify the issue pointed out by the Reviewer to read: "Additionally, significantly higher *induction of Myd88, Il1 α , Il1 β , and Tnfa* was evident in LPS TRAP samples but not in LPS input or negative fractions relative to PBS treatment (Supplementary Figure 18)."

11. In figure 8 there is a discrepancy between DNA methylation and gene expression. Can that be explained by the timing of sample collection? Or is there another good explanation

Response: At the positive fraction level (microglia), there seems to be correlation between hypomethylation of gene promoter by BSAS and enhanced transcription by RNAseq for all transcripts shown in response to LPS treatment. We interpret our findings to suggest that at the input level (which includes all cell types), microglia-specific epigenomic methylation/demethylation responses to LPS are diluted and no change in % mCG at the promoter region of the target genes is observed. However, other mechanisms of gene regulation in response to acute LPS administration might also take place that explain up-regulation of those genes in the absence of promoter methylation differences, which is noted in the text.

The timing of sample collection is the same for BSAS and RNA-seq, the same mice were used for TRAP and INTACT protocols, just different hemispheres, and subsequent processing and analyses.

12. For figure 8, some more genes should be analyzed. Including genes that do not show a response to LPS. And the expression values and methylation levels should be correlated. Genome-wide is perhaps not necessary, but 3 genes is very limited to draw robust conclusions.

Response: We invite the Reviewer to evaluate Supplementary Figure 19. The genomic methylation and RNA expression levels for *Gpr84*, *Aif1*, *Fabp7*, and *Eno2* are shown that complement the data on Figure 9 (former Figure 8 in the first submission).

13. In the discussion cell-specific contributions to the transcriptome and epigenome are mentioned. However, single cell RNA-sequencing data from recent years have shown that astrocytes and microglia are not homogeneous cell populations. Do the authors see a contribution for their technology (possible modified) for that?

Response: This is certainly an area we have planned for future investigation in which the GFP and mCherry labels could be used for intact cell isolations. While the single cell technologies are opening new doors to transcriptomic and epigenomic analysis they are not without caveats. The number of genes observed in single cell droplet transcriptome analysis is often only ~2,000-3,000 per cell and the expression measurements are 3' biased. This leaves a shallow coverage of the transcriptome and most differential RNA splicing unexplored. The process to isolate cells, typically FACS, can also cause activational artifacts (see Haimon et. al)⁹. We have added to the discussion that this should be a future area of investigation.

Minor comments

1. Genes and gene products (including RNA molecules) need to be written in italic (now it is sometimes confusing since also proteins are described in the text)

Response: We appreciate the observation by the Reviewer. The new version of this article shows gene names written in italic in the text, figures, and figure legends.

2. There are quite a few small mistakes /typo's (some examples: line 58, microglial: the I should be deleted, line 109: studes should read studies, line 123: putatively should read putative, etc.)

Response: We appreciate the observation by the Reviewer. The present version of this work has been carefully revised and mistakes/typos corrected.

3. Figure 2C, D and 5C, D the labeling (spelling) of categories on the x-axis of the violin and next to the heatmaps are inconsistent

Response: The updated versions of both Figures 2 and 5 have been corrected to make such labeling of categories consistent.

4. In the methods, the *vav*-Cre is mentioned. And that it did not work, because that particular cre driver is not very specific, also not in these essays. This information is hidden away now, but if the authors

think this is an important point to make, maybe this should also be briefly mentioned in the results section. Otherwise I would recommend to delete it

Response: We have removed the information about the Vav1-cre mouse line from the manuscript.

Reviewer #2 (Remarks to the Author):

Major Issues:

1. The TRAP negative fraction is not truly a negative fraction, but rather a positive-**depleted fraction**. **It's** impossible to remove all positive nuclei with TRAP, or INTACT for that matter, and thus impossible to **generate a true "negative" fraction**. **One would** need to FACS and gate for the mCherry-negative population to obtain a true negative fraction. Call the negative fraction a depleted fraction, or something to that effect.

Response: We agree with the Reviewer in that the protocols used in this study do not result in 100% yield in the positive fraction. However, to simplify the delivery of the fundamentals of the isolation methods [the idea that if there is a positive fraction, a negative (relative to the positive) fraction is left behind], we used standard terminology to call the TRAP/INTACT enriched fraction as "positive fraction" and depleted fraction as "negative fraction". Nonetheless, we understand the reviewer's point and have added information in Methods describing this caveat.

2. In figure 3 the authors show that ~90% of mcherry (astrocyte)-stained nuclei are bound to streptavidin. This results speaks to high affinity of streptavidin for biotinylated astrocyte nuclei, but the purification of the nuclei population in the positive fraction is unclear. Isolated nuclei are notoriously sticky, which raises the possibility that the NuTRAP/INTACT-selected astrocyte nuclei might be coupled to non-astrocyte nuclei. This would contribute to some degree of contamination. The authors should measure the percentage of mCherry-negative/strept. bead-**bound nuclei in the "positive" fraction in figure 3A and 3B**.

Response: Our apologies for any confusion here. To assess the purity of the nuclei in the positive fraction we followed the approach from the original Roh et al.¹ report. In the input fraction the number of mCherry+ nuclei is measured. In the positive fraction the mCherry signal overlaps with the streptavidin autofluorescence. Thus, the number of nuclei with streptavidin bound to them in the positive fraction is ~90% with only the remaining small fraction potential 'hitchhiker' nuclei.

3. As mentioned above, the impact of the biological insights generated from this manuscript are not immediately clear.

Response: The Discussion section in the present version of our manuscript expands on the biological insights generated in our work, especially in regards to DNA modifications in different cell types.

Minor Issues:

1. Show the representative tracks for the WGoXBS data.

Response: In the present version of our manuscript we show newly acquired BSAS data to address the Reviewer's request. This is much higher depth data than the whole genome analyses and therefore provides high confidence quantitation of CG and CH modifications. We added Figure 8 and Supplementary Figure 15 which give base-specific mCG and mCH comparisons for selected astrocytic and microglial genes between input, Aldh111-NuTRAP positive fraction, and Cx3cr1-NuTRAP positive fraction. Included in Figure 8 are schematics of the location of amplicons generated for bisulfite sequencing with respect to the transcription start site and exons

of the astrocytic and microglial marker genes. Of note is the fact that each region seems to have a distinct topography of mCG and mCH which is shifted to either higher or lower levels of methylation by fraction.

2. Show the head maps for the LPS RNA-seq data.

Response: The heatmap for the LPS RNA-seq data is shown in Figure 9c (former figure 8c).

3. Show Principle Components Analysis (PCA) for the WGoXBS experiments.

Response: A PCA of the WGoXBS data has been added to Figure 7e.

Reviewer #3 (Remarks to the Author):

1) The sentence beginning on line 200 claims: “Moreover, no studies have compared the methylome of different cell types, such as astrocytes and microglia, using a combination of inducible cre-recombinase **and NuTRAP technologies.**” Also, in the sentence beginning on 331 claims: “**In summary, the results** offer extensive evidence to support the combination of inducible cre/lox and NuTRAP models as a suitable and powerful approach for the parallel study of the cell-specific epigenetic and transcriptomic **signatures in the brain.**” Also, line 383: “**Collectively,** our experiments demonstrated that the NuTRAP **approach can be applied to CNS cell populations and...**” There are two issues with these and similar claims.

First, these claims could leave readers with the impression that this manuscript represents the first report combining epigenetic and transcriptomic signatures of specific cell types of the mouse brain. In fact, work with the original TRAP (Mellen et al., 2012) as well as INTACT (Mo et al., 2015, yes DNA and RNA were studied) and Tagger (Kaczmarczyk et al., 2019) all did this previously. The manuscript would be greatly improved if the claims of technical novelty were replaced with a comparison of all relevant published methods and with a focus on the new biological data, specifically the methylation studies.

Response: We appreciate the concern of the Reviewer in clarifying the message of novelty of our work. As expressed in the original version of our manuscript, we continue here to emphasize that the use of inducible cre/lox technology in combination with NuTRAP models as a suitable and powerful approach for the parallel study of the cell-specific epigenetic and transcriptomic signatures in the brain. The authors recognize the contributions of Mellen et al.,¹⁰ in demonstrating a cell-specific relationship between the genomic distribution of 5hmC, 5mC and gene expression in Purkinje cells and granule cells through a combination of 5hmC pulldown/MeDIP sequencing on sorted nuclei and TRAP-Seq, respectively. Similarly, the authors acknowledge and cite the work of Mo et al.,¹¹ in which nuclei were also sorted prior to epigenomic analyses of subpopulations of neurons of interest. Apart from the specific cell types that are subject of our study, the points that technically differentiates our study from Mellen, Mo, and others in the field are: 1) the temporal control of cre-recombination: It has been long recognized that successful gene targeting with cre and cre/ERT2-loxP is critically determined by the precise expression pattern of cre recombinase. A single cre-mediated excision event is passed on during cell division to all subsequent daughter cells. Therefore, brief expression of cre in an unreported cell type early in development can have profound consequences on any later phenotype or fate-mapping experiment.¹² Such issue is especially important to be overcome in epigenetic/transcriptomic studies of the aging brain, a central interest of our research group. Thus, validated models that allow short, temporary activation after development rather than permanent activation of cre recombinase for the study of epigenetic and transcriptomic landscapes of the brain would be of benefit to the field. The Discussion section of the new version of this article continues to indicate the value of Tamoxifen inducible cre in regards to avoiding off-target recombination during development and potential confounds of the downstream analyses. A novel application, that we are at the moment exploring, is the use of cre/ERT2; NuTRAP not only to tag cell-specific nuclei and polysomes but also to delete genes of interest in the cells being tagged, in the presence of a target floxed allele; 2) direct application of the fundamentals of the transgenic construct by affinity purification methods (INTACT) while avoiding tedious and long nuclei

sorting protocols for the purpose of cell-specific gDNA isolation for epigenomic studies; 3) TRAP and INTACT are present in the same construct: the population of cells evaluated by TRAP-RNAseq and INTACT-WGoxBS are consistent, since both tags localize to the same cell.

Regarding the request of the Reviewer of a comparison of all relevant published methods and with a focus on the new biological data, specifically the methylation studies: Extensive comparisons are offered along the present manuscript comparing TRAP data from this study with RiboTag models using the same or different cre-reporter lines to study the transcriptome of astrocytes (Figure 2) and microglia (Figure 5). For the methylation studies, such comparisons are not possible as no published study has yet reported astrocyte-specific or microglia-specific whole-genome oxidative sequencing or bisulfite oxidative sequencing data to determine microglia- and astrocyte-specific levels of mCG, mCH, and mhCG. Our DNA modification approaches provide base or region-specific absolute quantitation while prior studies utilized pulldown approaches which produce peak data. Combining these different types of data is a challenge given that they are fundamentally different.

2) Second, please explain the value of the temporal control for NuTRAP and why this is an important development that warrants being repeated throughout the manuscript. What is gained by turning the system on after development? Is the construct toxic if expressed during development? Probably not. The authors write that it might be detrimental to express cre through development but they offer no evidence that this is a problem that requires limited cre activity, not for CNS cells including the glia types studied in this report. The only apparent need for the demonstrated experiments is that it reduces the activation of off-target cells. This is why Cre-ERT2 lines are typically used for glia. In other words, the value in using a tamoxifen Cre line was to control the off targeting of cells by the Cre (the shortcomings of Cre) not due to a problem with constitutive expression of NuTRAP. A better explanation of the value of the temporal control as it relates to the current data is needed. Otherwise, this is simply a technical demonstration that the system can be induced and the text should be accordingly changed. In this case, the main result was that the system was employed to study brain glia. A similar mouse line expressing multiple proteins to study gene expression called Tagger was used to study brain neurons, so the ability to study glia is not unexpected.

Response: As expressed above, the temporal control is especially important in epigenetic/transcriptomic studies of the aging brain, an emerging direction in neuroscience and a central interest of our research as a group. The application of temporally induced recombination in this case would also be of benefit since short, temporary activation rather than permanent activation of cre recombinase is preferred. The Discussion section of the new version of this article continues to indicate the value of Tamoxifen inducible cre in regards to avoiding off-target recombination during development and potential confounds of the downstream analyses. A novel application, not discussed in this article, that we are at the moment exploring, is the use of cre/ERT2; NuTRAP not only to tag cell-specific nuclei and polysomes but also to delete genes of interest in the cells being tagged, in the presence of a floxed allele of interest. To this end, temporally controlled cre recombination is preferred.

While the purpose of our study is not the demonstration of the need of inducible cre-NuTRAP models as opposed to constitutive cre-NuTRAP models, the experimental design we chose and offer to the scientific community aims to circumvent potential confounds in the study of adult and aging cell populations, without the permanent off-target labeling that might occur during development, and minimize the exposure of the target cells of our studies to active cre and its products.

3) Another problem is the phrase **“paired analysis”** implies that both types of data are generated from the same sample. However, it is apparent that INTACT and TRAP were not done on the same samples since the homogenates were made with different buffers. Perhaps one hemisphere was used for one method and **the second hemisphere was used for the other method? The phrase “in parallel” was used** in places but this does not mean they were from the same brain. That should be clarified for the interpretation of the presented data. Also, since different buffers were used for the different methods, the workflows in supplemental figures 1 and 10 are very misleading and must be corrected. They imply that the same homogenate was split for the different methods but this is not true.

Response: As requested by the reviewer, the summary cartoon representing the methodology workflow (Supplementary figure 1) has been now modified for consistency with the described methods in the present version of this manuscript.

4) More importantly, the authors are obtaining only one level of gene expression from each lysate. The method would be much more useful if getting multiple domains from the same homogenate. This was accomplished with the other methods. Nuclei and TRAP should be done on the same homogenate, or at the very least a protocol should be established where that could be done. If this was attempted but without success then that information should be clearly noted in a prominent location (e.g. discussion) of the manuscript. This is the most valuable purpose for these types of models (NuTRAP and Tagger) and readers will undoubtedly want to know this.

Response: We are puzzled by the perception of the Reviewer that the method would be much more useful if getting multiple domains from the same homogenate, and as such we have no response on that point.

During the preparation of this manuscript, the authors attempted the purification of nuclei and polysomes from a single brain homogenate. Because the buffer composition that is critical for isolation of good quality nuclei and polysomes does not completely overlap, and the yield and purity of TRAP RNA was significantly diminished when secondary to nuclei separation, the authors chose to work with parallel samples from the same brain sample, that is one brain hemisphere for TRAP and one hemisphere for INTACT. We honestly believe it is at the readership's discretion to apply the models presented in this manuscript to achieve more practical outcomes, such as TRAP- and INTACT- sequencing from the same sample preparation that might continue to benefit the field.

5) NGS samples were purified from hemibrains but the histology only shows hippocampal areas. The inclusion of images of the whole brain should therefore be displayed. Any images that include only a **specific region (e.g. figures 1c and 4c) should have that region clearly labeled (“Brain” is not a sufficient label.)** Readers will want to know if the activation driven by these cre lines worked as well for NuTRAP as they did for RiboTag. Any differences would be useful information to researchers designing experiments, as well as readers of this area of the scientific literature.

Response: Figures 1 and 4 will continue to show representative images of the hippocampus (now described in the respective figure legends) for all markers used. For the present resubmission we have taken additional confocal images of brain regions (cortex and cerebellum), along with hippocampus for both models that are included as Supplementary Figure 2 (for the Aldh111-NuTRAP) and Supplementary Figure 6 (for the Cx3cr1-NuTRAP) to show co-localization of mCherry and EGFP in recombined cells. We have also captured high resolution tile scan images covering the entirety of a sagittal brain section for both mouse lines to further depict the overlap of eGFP and mCherry and extent of recombination upon Tam induction. Images are available through two private links (<https://figshare.com/s/fc817f9235f4384c5fb8> for Aldh111-NuTRAP and <https://figshare.com/s/de4f6fcb1c8fc858de10> for Cx3cr1-NuTRAP). The present Methods section reflects the details of the new methodology and software used for data acquisition. As well, hippocampus specific TRAP data is now included in the supplement for comparison in the Cx3cr1-NuTRAP model.

6) The FACS studies applied gating windows that excluded certain populations of cells that appeared to be GFP+ (dots above the gate windows in the main text figures). It seems reasonable that these cells were expressing GFP and TRAP, and would therefore be in the pool of captured nuclei and RNA. Indeed, **in supplemental figure 4 B a” there is prominent GFP staining of a cell sized and shaped more like a neuron than microglia.** What was the rationale for not including an analysis with all the GFP+ cells in the FACS analysis? That is, how were the dots above the gating windows excluded? Furthermore, why not also do an experiment where the first gate is for the cell marker (e.g. cd11b) then determine how many of those are expressing GFP? It seems the purpose of the FACS study should have been to determine the completeness and specificity of the proportion of GFP+ cells in the different glia, but the data are not

presented in a way that addresses these important questions. Also, a definition of PE on the Y-axis should be included.

Response: In our hands, by flow cytometry (FC), the intrinsic signal of EGFP upon Tamoxifen treatment was consistently detected in brain cell suspensions while the fluorescence associated to mCherry signal was not. This observation was similar for immunohistochemistry of brain sections, by which mCherry expression was successfully detected and co-localized to EGFP-expressing cells through staining with anti-mCherry antibody but not by intrinsic fluorescence as for EGFP. Only in fresh nuclear preparations, the intrinsic expression of mCherry was observed without antibody staining (see input sample in Figures 3a and 6a). Considering the detection caveat for mCherry fluorescence detection, for the purpose of assessment of cell-specificity of cre-recombination, EGFP expression was the criteria followed to gate on the population of recombined cells. The gating strategy was selected upon consideration of all proper controls (Supplementary Figures 21 and 22) and color compensation to clearly depict the cell population of interest with minimal background in both cre-positive and cre-negative samples. Any ungated cells above the gating window, are within a region excluded for analysis in all experimental samples (they represent autofluorescence), as per the nature of this type of analysis. As defined in the Methods section, a 488 nm (blue) laser with 530/30 and 580/30 filter combinations was used to gate on EGFP⁺ cells within single cells (singlets) without auto-fluorescence interference. Since the 530/30 (PE) filter was used to gate-out autofluorescence in our strategy, no additional labeling in the Y-axis is provided.

The analysis of EGFP⁺ cells on gated CD11b⁻ expressing cells was done in the same experiments shown in Figures 1 and 3 and is now shown in Supplementary Figures 23, 24, and 25 in the present version of our manuscript.

References:

1. Roh, H.C. *et al.* Simultaneous Transcriptional and Epigenomic Profiling from Specific Cell Types within Heterogeneous Tissues In Vivo. *Cell Rep* **18**, 1048-1061 (2017).
2. Sun, W. *et al.* SOX9 Is an Astrocyte-Specific Nuclear Marker in the Adult Brain Outside the Neurogenic Regions. *J Neurosci* **37**, 4493-4507 (2017).
3. Verkhratsky, A. & Nedergaard, M. Physiology of Astroglia. *Physiol Rev* **98**, 239-389 (2018).
4. Aguzzi, A., Barres, B.A. & Bennett, M.L. Microglia: scapegoat, saboteur, or something else? *Science* **339**, 156-161 (2013).
5. Boisvert, M.M., Erikson, G.A., Shokhirev, M.N. & Allen, N.J. The Aging Astrocyte Transcriptome from Multiple Regions of the Mouse Brain. *Cell Rep* **22**, 269-285 (2018).
6. Itoh, N. *et al.* Cell-specific and region-specific transcriptomics in the multiple sclerosis model: Focus on astrocytes. *Proc Natl Acad Sci U S A* **115**, E302-E309 (2018).
7. Srinivasan, R. *et al.* New Transgenic Mouse Lines for Selectively Targeting Astrocytes and Studying Calcium Signals in Astrocyte Processes In Situ and In Vivo. *Neuron* **92**, 1181-1195 (2016).
8. McKenzie, A.T. *et al.* Brain Cell Type Specific Gene Expression and Co-expression Network Architectures. *Sci Rep* **8**, 8868 (2018).
9. Haimon, Z. *et al.* Re-evaluating microglia expression profiles using RiboTag and cell isolation strategies. *Nat Immunol* **19**, 636-644 (2018).
10. Mellen, M., Ayata, P., Dewell, S., Kriaucionis, S. & Heintz, N. MeCP2 binds to 5hmC enriched within active genes and accessible chromatin in the nervous system. *Cell* **151**, 1417-1430 (2012).
11. Mo, A. *et al.* Epigenomic Signatures of Neuronal Diversity in the Mammalian Brain. *Neuron* **86**, 1369-1384 (2015).
12. Chow, L.M., Zhang, J. & Baker, S.J. Inducible Cre recombinase activity in mouse mature astrocytes and adult neural precursor cells. *Transgenic Res* **17**, 919-928 (2008).

Reviewers' Comments:

Reviewer #1:

Remarks to the Author:

I appreciate the efforts made by the authors to respond to my comments and can now recommend publication of the manuscript

Reviewer #3:

Remarks to the Author:

In response to question 2, the authors answer that the tamoxifen induction is novel and will be important for aging research. If that is the story then present data from aged mice. As it stands the argument is unconvincing. The experiments are so completely obvious, no one would be surprised by this. This is not novel. So why spin it so much?

For question 3, they continue to widely use the phrase "paired analysis" which I find misleading, and they do not address this concern at all. It should be more clearly stated that "paired" can only be done if using one hemisphere for nuclei and the other for mRNA.

In response to question 4, the authors report being puzzled by the suggestion that it would be valuable to obtain multiple fractions from the same lysate. I find this hard to believe.

The authors report data obtained from hemibrains. Imagine one is studying a specific region, such as the hippocampus, and both sides are needed for sufficient yields for TRAP. In such a case the ability to study the nuclei prepared from the same lysate would be useful. Is this a realistic scenario? According to the methods section the authors worked with isolated hippocampi but do not report the results. Why not? What was the yield? What was the yield when a hemibrain was used? These details should be reported so readers can determine if this is a method worth trying for their projects. It seems likely that the yield is too low when using a hippocampal isolate from one hemisphere. It's incredulous that the authors haven't thought that obtaining both from the same lysate would be useful. Nonetheless, now that such an idea has been seeded, and they admit having tried but failed, they must mention this. Most experimentalists wanting to apply these methods would want to obtain both from the same lysate. Not being able to also purify nuclei from the same lysate as TRAP is a problem and it is not fair to readers, especially those who may consider using this method, to hide this information.

The response to question 6 does not clarify the issue. More bluntly, what percent of CD1 positive cells are captured and what does this mean for the bioinformatics analysis and the method in general? Also, the point about gating-out autofluorescence should be more clear. Some readers would not expect to see such a wide range, and it may impact on studies to capture glial cells that are dimly fluorescent.

In summary, if the authors already have information that there are difficulties with their methods they should warn researchers who might be interested in applying these methods. Since the authors explain that this manuscript is essentially a validation study, it defies logic that trouble spots should be hidden. The authors emphasize in their rebuttal that the main novelty of their experiments, where they cross mouse lines they bought, is that they used tamoxifen for induction. Unsurprisingly, it worked. That is only a small step forward. My enthusiasm expressed during the initial review is seriously diminished.

Reviewer #3 Comments to Authors

In response to question 2, the authors answer that the tamoxifen induction is novel and will be important for aging research. If that is the story then present data from aged mice. As it stands the argument is unconvincing. The experiments are so completely obvious, no one would be surprised by this. This is not novel. So why spin it so much?

We disagree with the critique that these results are not novel. The reviewer is conflating novelty and what they call obvious. However, this is both poor logic and not supported by their argument. There is no refutation that these experiments are novel and that the findings provide deeper insight into the molecular profiles, especially in DNA modifications, of different CNS cell types and this model system could be of great utility in neuroscience studies. What specific spin are they referring to and what results are unsurprising? As primarily a technical report that the model works as one would hope should be a strength not a vague weakness. The desire for aging studies, which are ongoing but would require two years of aging mice is far outside of the scope of this report and an unreasonable request.

For question 3, they continue to widely use the phrase “paired analysis” which I find misleading, and they do not address this concern at all. It should be more clearly stated that “paired” can only be done if using one hemisphere for nuclei and the other for mRNA.

In the revision a clear diagram (Figure 1) of how the experiments are performed and the model is used is presented along with additional text. Paired refers to how these TRAP and INTACT isolations are performed as a pair, in other words, two similar things. In statistics, “paired” samples are samples in which natural or matched couplings occur. Having paired data from the same animal allows for paired statistical analyses and correlations to be done. Much as one has a pair of shoes or pair of gloves you have two separate but similar or parallel things. We would argue the reviewer is defining ‘paired’ as synonymous with ‘simultaneous’, which is not the standard definition.

In response to question 4, the authors report being puzzled by the suggestion that it would be valuable to obtain multiple fractions from the same lysate. I find this hard to believe.

The authors report data obtained from hemibrains. Imagine one is studying a specific region, such as the hippocampus, and both sides are needed for sufficient yields for TRAP. In such a case the ability to study the nuclei prepared from the same lysate would be useful. Is this a realistic scenario? According to the methods section the authors worked with isolated hippocampi but do not report the results. Why not? What was the yield? What was the yield when a hemibrain was used? These details should be reported so readers can determine if this is a method worth trying for their projects. It seems likely that the yield is too low when using a *hippocampal isolate from one hemisphere. It’s incredulous that the authors haven’t thought that obtaining both from the same lysate would be useful.* Nonetheless, now that such an idea has been seeded, and they admit having tried but failed, they must mention this. Most experimentalists wanting to apply these methods would want to obtain both from the same

lysate. Not being able to also purify nuclei from the same lysate as TRAP is a problem and it is not fair to readers, especially those who may consider using this method, to hide this information.

The hippocampal TRAP data is shown in Figure 7. The broader point is that future investigations will have to determine if a single hemisphere of a bilateral structure or both hemispheres are needed. Brain regions vary a great deal in size and cellular content and investigators will have to determine the nucleic acid yield after isolation of their region of interest is sufficient to achieve their experimental goals. Previous papers that were highlighted by reviewer 3 (NuTRAP and Tagger, first review of this manuscript) also used a paired, parallel approach where nuclei and polysomes/RNA are isolated from distinct tissue from the same animal.

The response to question 6 does not clarify the issue. More bluntly, what percent of CD1 positive cells are captured and what does this mean for the bioinformatics analysis and the method in general? Also, the point about gating-out autofluorescence should be more clear. Some readers would not expect to see such a wide range, and it may impact on studies to capture glial cells that are dimly fluorescent.

It is unclear if the reviewer is questioning the efficiency of the Cx3cr1 line cre recombination or the percentage of Cd11b+ cells that have EGFP (this is in Supplemental Figure 25). More broadly these flow cytometry figures are cell analyses to provide a phenotype of the cells labeled with the NuTRAP construct and not cell sorting for subsequent nucleic acid isolation (the very approach this methodology improves upon). The question of auto-fluorescence seems to be conflated with staining controls. The extensive controls (antibody and isotype controls, and unstrained cells from cre+ and cre- mice) for the flow cytometry are provided in Supplementary Figure 21-24 and further details are provided in the flow cytometry methods. We are unable to discern the meaning of what is written about how the flow cytometry is related to the bioinformatics.

In summary, if the authors already have information that there are difficulties with their methods they should warn researchers who might be interested in applying these methods. Since the authors explain that this manuscript is essentially a validation study, it defies logic that trouble spots should be hidden. The authors emphasize in their rebuttal that the main novelty of their experiments, where they cross mouse lines they bought, is that they used tamoxifen for induction. Unsurprisingly, it worked. That is only a small step forward. My enthusiasm expressed during the initial review is seriously diminished.

All of the molecular data is publically deposited, reams of supplemental data, and extensive methods are provided in full transparency. The manuscript also describes in extensive detail how we have obtained reproducible results across many sets of mice and different endpoints.